# Don't Force the Fit: Bounded Log-Likelihood Loss for Enhanced Reasoning in Large Language Models

**Feng Zhao** [1]   **Hong Zhang** [1]   **Yu Yang** [2]   **Ruilin Zhao** [1]   **Guandong Xu** [2]

## Abstract

Supervised fine-tuning (SFT) is central to aligning large language models (LLMs) with instruction following and task-specific reasoning. Despite its success, SFT optimizes token-level likelihoods under the implicit assumption that strictly fitting all tokens in expert demonstrations induces the desired downstream behavior. However, in reasoning tasks where correctness is defined by logical validity or final outcomes rather than exact token realizations, this assumption can lead to optimization misalignment. We empirically observe that low-probability tokens in reasoning demonstrations often correspond to realization-specific or stylistic variations, and that reducing their influence during training consistently improves generalization on reasoning benchmarks. Motivated by this insight, we propose the *Bounded Log-Likelihood Loss* (BLL-Loss), a simple and parameter-free alternative to standard likelihood training that bounds gradient contributions from low-probability tokens while preserving conventional optimization behavior. We provide theoretical insights and extensive empirical results demonstrating that BLL-Loss improves reasoning generalization across diverse model scales and challenging benchmarks.

## 1. Introduction

Supervised fine-tuning (SFT) is central to aligning large language models (LLMs) with reliable instruction following and task-specific reasoning (Wei et al., 2022; Ouyang et al., 2022). Despite its widespread success, SFT is often treated as a purely behavioral alignment step, with limited systematic examination of whether its token-level optimization objectives are fundamentally aligned with the requirements of reasoning generalization (Grattafiori et al., 2024; Shao et al., 2025; Chen et al., 2025).

Standard SFT employs a token-level likelihood objective that encourages models to assign high probability to every token in expert demonstrations, implicitly assuming that faithfully fitting the empirical data distribution suffices to induce the desired behavior (Wang et al., 2025; Ye et al., 2025). However, reasoning correctness is defined at the level of logical consistency or final outcomes, not surface token realizations (Mirzadeh et al., 2025). As logically equivalent reasoning can admit diverse syntactic forms, empirical data often contains stylistic and formatting variations irrelevant to the underlying reasoning (Wang et al., 2023; Zhou et al., 2023b; Wei et al., 2025), which token-level optimization may amplify and thereby hinder the learning of core reasoning structures.

This challenge is exacerbated in contemporary LLMs, which incorporate powerful inductive biases and linguistic priors via large-scale pre-training (Liu et al., 2025; Hu et al., 2025). With the standard likelihood objective, tokens that clash with these priors incur high losses, producing disproportionate gradient updates that force the model to distort its intrinsic structures to fit trivial surface realizations. This prompts a fundamental question: *Is strict token-level fitting necessary for reasoning alignment?*

In this work, we provide empirical evidence that strict token-level fitting is often unnecessary for effective reasoning alignment. Through empirical analysis, we find that low-probability tokens in reasoning demonstrations frequently correspond to stylistic or realization-specific artifacts rather than core reasoning steps. Surprisingly, suppressing the influence of such tokens during fine-tuning consistently improves reasoning performance across a wide range of benchmarks. These observations suggest that, in reasoning alignment, strict adherence to every demonstrated token may introduce optimization interference rather than informative supervision.

Motivated by this insight, we propose the *Bounded Log-*

---

[1]Natural Language Processing and Knowledge Graph Lab, School of Computer Science and Technology, Huazhong University of Science and Technology, Wuhan, China [2]Centre for Learning, Teaching and Technology, The Education University of Hong Kong, Hong Kong SAR, China. Correspondence to: Feng Zhao <zhaof@hust.edu.cn>.

*Proceedings of the $43^{rd}$ International Conference on Machine Learning*, Seoul, South Korea. PMLR 306, 2026. Copyright 2026 by the author(s).

*Likelihood Loss* (BLL-Loss), a smooth and parameter-free alternative to standard likelihood training. BLL-Loss selectively limits the influence of low-probability tokens while preserving standard optimization behavior on high-confidence regions, thereby aligning supervised fine-tuning with the structural properties of reasoning tasks. Importantly, BLL-Loss is not an ad hoc modification. We show that it emerges naturally from two complementary theoretical perspectives, providing a principled justification for bounding gradient contributions from conflicting tokens. Extensive empirical evaluations further demonstrate that BLL-Loss consistently improves reasoning generalization across diverse model scales and challenging benchmarks. More broadly, our results suggest that effective reasoning alignment does not require strictly fitting every token in expert demonstrations, but rather respecting the structural regularities already encoded by pre-trained models.

Our contributions are threefold:

- We provide a principled characterization of token-level optimization in reasoning alignment, showing via theory and diagnostics that strict per-token fitting is unnecessary and that bounding low-probability token gradients improves reasoning generalization.

- We propose the BLL-Loss, a simple, parameter-free alternative to standard likelihood training. BLL-Loss selectively bounds the influence of low-probability tokens, providing a principled mechanism for mitigating outlier-driven optimization effects.

- We empirically demonstrate that BLL-Loss consistently improves performance on challenging reasoning benchmarks across diverse model scales and architectures, demonstrating strong robustness and general applicability.

## 2. Empirical Diagnosis of SFT Optimization in Reasoning

### 2.1. SFT Objective and Gradient Sensitivity

SFT adapts a pre-trained policy $\pi_\theta$ to approximate the behavioral distribution of expert reasoners by maximizing the likelihood of curated demonstrations. Given a dataset $\mathcal{D} = \{(x_i, y_i)\}_{i=1}^{N}$, where $x_i$ denotes an input prompt and $y_i = (y_{i,1}, \ldots, y_{i,T_i})$ is the corresponding target sequence, SFT minimizes the negative log-likelihood (NLL):

$$\mathcal{L}_{\text{SFT}}(\theta) = \mathbb{E}_{(x,y)\sim\mathcal{D}} \left[ -\sum_{t=1}^{T} \log \pi_\theta(y_t \mid x, y_{<t}) \right]. \quad (1)$$

This objective is equivalent to minimizing the forward KL divergence $\text{KL}(\pi_E \parallel \pi_\theta)$ between the empirical expert dis-

tribution $\pi_E$ and the model distribution, corresponding to behavior cloning.

To analyze the optimization dynamics, we examine the gradient of the loss at the token level. Let $z_\theta(x, y_{<t}) \in \mathbb{R}^{|\mathcal{V}|}$ denote the pre-softmax logits at time step $t$, and let

$$\pi_\theta(k \mid x, y_{<t}) = \frac{\exp(z_k)}{\sum_j \exp(z_j)} \quad (2)$$

be the induced softmax distribution. The gradient of the token-level loss with respect to the logits is given by

$$\frac{\partial}{\partial z_k}\big[ -\log \pi_\theta(y_t \mid x, y_{<t}) \big] = \pi_\theta(k \mid x, y_{<t}) - \mathbb{I}[k = y_t], \quad (3)$$

where $\mathbb{I}[\cdot]$ denotes the indicator function.

Equation (3) implies that the gradient on the target logit is $\frac{\partial}{\partial z_{y_t}}[-\log \pi_\theta(y_t \mid x, y_{<t})] = \pi_\theta(y_t \mid x, y_{<t}) - 1 = -(1 - \pi_\theta(y_t \mid x, y_{<t}))$, which is bounded in $(-1, 0)$ and approaches $-1$ when the model assigns low probability to the target token. For non-target logits $z_k$ with $k \neq y_t$, the gradient is $\frac{\partial}{\partial z_k}[-\log \pi_\theta(y_t \mid x, y_{<t})] = \pi_\theta(k \mid x, y_{<t}) \geq 0$. Therefore, any deviation from the demonstrated trajectory induces a corrective update that uniformly favors the observed token. Since this token-level signal depends solely on exact symbol matching, the SFT objective implicitly treats all mismatches as equally significant, regardless of whether they reflect a logically essential reasoning step or a realization-specific variation among multiple valid solutions.

### 2.2. Problem Formulation: Optimization vs. Reasoning

> **Key Takeaway:** The optimization signal produced by standard SFT operates purely at the token level and does not account for the semantic role that individual tokens play in a reasoning process.

Standard SFT is typically interpreted under a knowledge injection paradigm, in which high-loss tokens are regarded as signals of missing capabilities and are therefore subjected to strong corrective updates. However, recent work on the superficial alignment hypothesis (Zhou et al., 2023a) suggests that the core knowledge of modern language models is largely acquired during pre-training, with subsequent fine-tuning primarily serving to elicit and expose existing capabilities. This perspective is particularly important in reasoning tasks. Unlike factual memorization or closed-form prediction, reasoning problems often admit multiple equivalent but differently expressed intermediate reasoning paths under a logically unique conclusion. Consequently, token-level deviations in reasoning settings do not necessarily indicate deficiencies in capability.

To formalize this distinction, let $h_t = (x, y_{<t})$ denote the

decoding history and $\mathcal{E}(x)$ the equivalence class of logically valid solutions to $x$. We define $\mathcal{V}_{\text{valid}}(h_t) \subset \mathcal{V}$ as the set of next tokens that can extend $h_t$ into $\mathcal{E}(x)$, and define the *logic confidence* as

$$\rho(h_t) = \sum_{v \in \mathcal{V}_{\text{valid}}(h_t)} \pi_\theta(v \mid h_t).$$

For a demonstrated token $y_t \in \mathcal{V}_{\text{valid}}(h_t)$, high surprisal arises either when $\rho(h_t) \ll 1$, indicating a genuine capability gap, or when $\rho(h_t)$ is large but $\pi_\theta(y_t \mid h_t) \ll \max_{v \in \mathcal{V}_{\text{valid}}(h_t)} \pi_\theta(v \mid h_t)$, indicating a realization-level mismatch. This ambiguity explains why token-level loss alone is insufficient to diagnose reasoning deficiencies. To address this, we explicitly define the scope of our research. In this work, we operate strictly within the capability elicitation paradigm rather than knowledge injection. We do not attempt to resolve genuine capability gaps. Instead, we focus exclusively on mitigating realization-level mismatches.

### 2.3. Investigation I: Token Likelihood Analysis of Reasoning Demonstrations

> **Key Takeaway:** The pre-trained prior already aligns well with the vast majority of reasoning-relevant tokens, while a sparse low-probability tail is dominated by task-irrelevant stylistic artifacts.

**Motivation.** The formulation above suggests that low-probability tokens in reasoning tasks can arise from two fundamentally different sources: genuine reasoning failures, characterized by low logic confidence $\rho(h_t)$, and realization-level mismatches (e.g., stylistic, narrative, or formatting variations), where the model assigns substantial probability mass to alternative valid continuations. To assess which regime dominates in practice, we empirically probe the quality of the pre-trained prior $\pi_{\theta_0}$ by analyzing token likelihoods on reasoning data.

**Experimental Setup.** We investigate the probability landscape of the NuminaMath-CoT[1] dataset using Qwen2.5-Math-1.5B (Yang et al., 2024). The Base model is selected to specifically isolate the pre-trained prior $\pi_{\theta_0}$ before any SFT-induced distortion. We compute conditional probabilities $p_t = \pi_{\theta_0}(y_t \mid x, y_{<t})$ over a sample of approximately $1 \times 10^6$ tokens. To interpret the semantic nature of these probabilities, we perform a stratified composition analysis, with detailed word cloud visualizations provided in Appendix D.1.

**Observation.** Figure 1 quantifies the resulting probability landscape, exhibiting a stark skewness that validates our hypothesis regarding model competence. The probability

[1]https://huggingface.co/datasets/AI-MO/NuminaMath-CoT

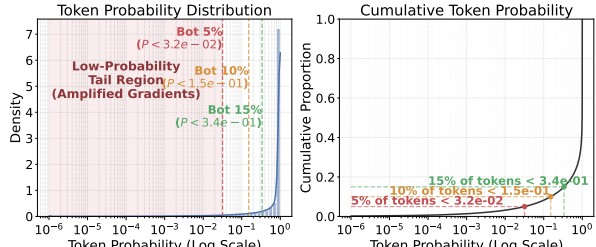

*Figure 1.* Token probability distribution in the SFT dataset. (Left) Log-scale distribution exhibiting a pronounced long tail. (Right) Cumulative proportion of tokens below each probability threshold.

mass heavily concentrates near $10^0$ (Regime of Competence), confirming that the base model already possesses sufficient prior knowledge. In stark contrast, a persistent heavy tail extends into the low-probability regime, where the bottom 5% of tokens fall below $3.2 \times 10^{-2}$ (and the lowest 10% below $1.5 \times 10^{-1}$). Furthermore, our qualitative analysis in Appendix D.1 reveals that this low-probability tail is disproportionately composed of task-irrelevant stylistic artifacts. This finding implies that low-probability tokens more often reflect realization-level mismatches rather than genuine gaps in reasoning capability.

### 2.4. Investigation II: Suppressing Low-Confidence Tokens in SFT Optimization

> **Key Takeaway:** Reasoning alignment does not require strict fitting of every token; masking low-probability tokens can instead lead to substantial improvements in reasoning performance.

**Motivation.** The analysis in the previous section shows that low-probability tokens are predominantly composed of stylistic or surface-level elements that are weakly related to core reasoning. In conjunction with Eq. (3), this implies that under standard SFT such tokens induce near-saturated gradient signals and are therefore repeatedly amplified during optimization. In reasoning tasks, this behavior reflects a trade-off: while strictly fitting these surface-level tokens improves distributional matching, excessive emphasis on signals weakly coupled to reasoning structure may consume limited representational capacity and interfere with the modeling of core reasoning patterns. Consequently, selectively suppressing gradient contributions from low-probability tokens may help release model capacity and improve generalization.

**Experimental Setup.** To empirically validate this, we conduct a controlled masking experiment. Using the cumulative distribution statistics derived from Investigation I (Section 2.3), we define three suppression thresholds corresponding to the bottom 5%, 10%, and 15% of the probability mass ($\tau \approx 0.032, 0.150, 0.340$, respectively). We train SFT

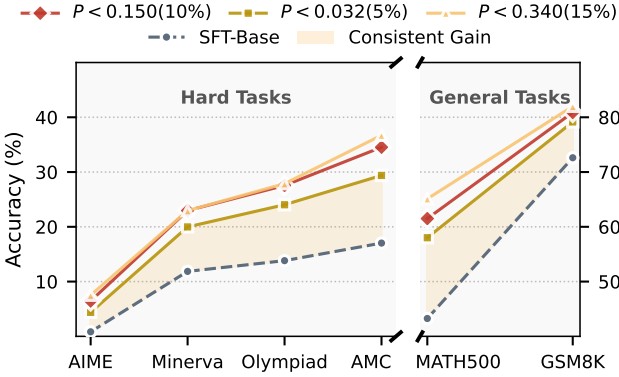

*Figure 2.* Performance impact of filtering low-probability tokens across hard and general reasoning tasks. The figure displays accuracy percentages on varied benchmarks, divided into "Hard Tasks" (left panel) and "General Tasks" (right panel).

variants of Qwen2.5-Math-1.5B where the loss for any token satisfying $\pi_\theta(y_t) < \tau$ is explicitly zeroed out, thereby preventing these high-surprisal stylistic artifacts from back-propagating large gradients.

**Observation.** Figure 2 shows that simply masking low-probability tokens consistently outperforms the standard SFT baseline across a wide range of benchmarks, with particularly pronounced improvements on challenging reasoning tasks such as AIME and Olympiad. This result reveals an optimization phenomenon that runs counter to conventional intuition: in reasoning alignment, not all tokens need to be strictly fitted. Low-probability tokens often introduce optimization interference, thereby undermining the learning of generalizable reasoning capabilities. This behavior stands in sharp contrast to optimization practices that emphasize greater focus on hard examples, such as Focal Loss (Lin et al., 2017).

**Reflection.** Although hard masking validates the intuition of suppressing stylistic tokens, it relies on discontinuous static thresholds and risks eliminating useful supervisory signals. As a result, hard masking is better viewed as a diagnostic tool rather than a final training strategy. To overcome these limitations and operationalize this intuition into a robust framework, we introduce the BLL-Loss in the next section. Furthermore, while our initial diagnostics focus on mathematical reasoning, mitigating over-optimization on low-confidence tokens is a domain-agnostic principle. We empirically verify the broad generalization of BLL-Loss on code and general reasoning tasks in Section 4.5.

# 3. Method: Bounded Log-Likelihood Loss (BLL-Loss)

While the hard masking experiments in Section 2.4 empirically demonstrate the benefit of suppressing low-probability

tokens, the approach itself constitutes an optimization compromise: reliance on discrete, heuristic thresholds introduces non-differentiable discontinuities into the loss landscape. To operationalize this principle within a fully differentiable framework, we propose the BLL-Loss. We derive BLL-Loss through two convergent theoretical lenses: **series convergence analysis** and **log-odds reparameterization**. Both perspectives lead to a unified, parameter-free objective whose gradient contribution is suppressed for low-probability tokens. For notational brevity in the following derivations, we denote the conditional probability of the target token as $\pi_\theta(y_t \mid x) := \pi_\theta(y_t \mid x, y_{<t})$.

## 3.1. First Perspective: Loss Design via Series Convergence

**Motivation.** To design a more robust loss, it is essential to carefully characterize the sensitivity of the cross-entropy (CE) loss to low-probability events. However, the closed-form expression $-\log \pi_\theta(y_t \mid x)$ alone provides little insight into the internal structure underlying this sensitivity. To make this mechanism explicit, we rewrite the CE loss as a power series in terms of $\epsilon = 1 - \pi_\theta(y_t \mid x)$:

$$\mathcal{L}_{\text{CE}} = -\log(1-\epsilon) = \sum_{n=1}^{\infty} \frac{\epsilon^n}{n} = \epsilon + \frac{\epsilon^2}{2} + \frac{\epsilon^3}{3} + \cdots. \quad (4)$$

This expansion reveals a critical structural flaw: $\mathcal{L}_{\text{CE}}$ is composed entirely of positive terms that accumulate constructively. As the error approaches the singularity ($\epsilon \to 1$, i.e., $\pi_\theta(y_t \mid x) \to 0$), the loss mimics the divergent behavior of the harmonic series. This explains why sparse outliers exert disproportionate influence: the unbounded accumulation of higher-order terms generates an overwhelming penalty signal that biases the optimization trajectory toward fitting the tail.

**Regularization via Alternating Series.** We seek a modification that attenuates this divergent tail behavior (robustness to noise) while preserving the linear response in the high-confidence regime (fidelity to signal). Inspired by the properties of the alternating harmonic series, where alternating signs are employed to regularize divergent expansions and improve convergence properties (Rudin, 1976), we construct a surrogate objective by imposing an alternating sign structure:

$$\mathcal{L}_{\text{new}} = \epsilon - \frac{\epsilon^2}{2} + \frac{\epsilon^3}{3} - \frac{\epsilon^4}{4} + \cdots. \quad (5)$$

This formulation exhibits a dual geometric profile. In the high-confidence regime ($\epsilon \to 0$), the objective is asymptotically linear ($\mathcal{L}_{\text{new}} \approx \epsilon$), strictly preserving the gradient dynamics of standard MLE. Conversely, in the low-confidence limit ($\epsilon \to 1$), the alternating series naturally saturates at a finite constant ($\log 2$). Unlike the unbounded divergence

of CE, this intrinsic boundedness neutralizes high-surprisal outliers without relying on heuristic clipping thresholds.

**The BLL-Loss Formulation.** Recognizing that the constructed series $\mathcal{L}_{\text{new}}$ is exactly the Taylor expansion of $\log(1 + \epsilon)$, we substitute $\epsilon = 1 - \pi_\theta(y_t \mid x)$ back to derive the closed-form BLL-Loss:

$$\mathcal{L}_{\text{BLL}} = \log\big(1 + (1 - \pi_\theta(y_t \mid x))\big) = \log\big(2 - \pi_\theta(y_t \mid x)\big). \tag{6}$$

For sequence generation, the sequence-level objective is defined as:

$$\mathcal{L}_{\text{BLL}} = \mathbb{E}_{(x,y) \sim \mathcal{D}} \left[ \sum_{t=1}^{|y|} \log\big(2 - \pi_\theta(y_t \mid x, y_{<t})\big) \right]. \tag{7}$$

Crucially, this result is not merely an artifact of series manipulation. In the following section, we demonstrate that this exact formulation can be independently derived from a completely distinct theoretical perspective—log-odds decomposition.

### 3.2. Second Perspective: Log-Odds Decomposition

**Motivation.** Recent advances in alignment, such as DPO (Rafailov et al., 2023), explicitly model optimization as maximizing the margin between preferred ($y_w$) and dispreferred ($y_l$) trajectories via a logistic objective:

$$\mathcal{L}_{\text{DPO}} = -\log \sigma\Big( \beta \log \frac{\pi_\theta(y_w \mid x)}{\pi_{\text{ref}}(y_w \mid x)} - \beta \log \frac{\pi_\theta(y_l \mid x)}{\pi_{\text{ref}}(y_l \mid x)} \Big). \tag{8}$$

While SFT lacks the explicit negative pairs typical of preference optimization, it inherently enforces a *latent contrastive mechanism* via the probability simplex: the target token $y_t$ competes against the aggregated probability mass of all other tokens in the vocabulary, i.e., $1 - \pi_\theta(y_t \mid x)$. We formalize this intuition by performing an algebraic decomposition (detailed in Appendix A) that recasts the standard Cross-Entropy loss into a form structurally analogous to Eq. (8):

$$\mathcal{L}_{\text{CE}} = -\log \sigma\Big( \underbrace{\log \pi_\theta(y_t \mid x)}_{\text{Target Promotion}} - \underbrace{\log\big(1 - \pi_\theta(y_t \mid x)\big)}_{\text{Complement Suppression}} \Big). \tag{9}$$

**Decoupling via the $\beta$-Balanced Framework.** Equation (9) expresses the per-token CE loss as a logistic form that combines two log-terms with distinct mathematical roles: *Target Promotion* ($\log \pi_\theta(y_t \mid x)$), which encourages agreement with the demonstrated token, and *Complement Suppression* ($\log(1 - \pi_\theta(y_t \mid x))$), which penalizes implausible alternatives. In standard MLE, these two components are rigidly coupled with equal weight. We re-parameterize the

relative balance of this objective at the loss level by introducing a balancing coefficient $\beta \in [0, 1]$ to control their relative contributions, thereby decoupling the strength of target matching from the strength of alternative suppression.

$$\mathcal{L}_\beta = $$
$$-\log \sigma\Big( \underbrace{\beta \log \pi_\theta(y_t \mid x)}_{\text{promotes target token}} - \underbrace{(1 - \beta)\log(1 - \pi_\theta(y_t \mid x))}_{\text{suppresses others}} \Big). \tag{10}$$

**Gradient Detachment and Unification.** Analysis of the gradient dynamics reveals that the limit $\beta \to 0$ induces a critical gradient detachment mechanism. By asymptotically eliminating the unbounded promotion term, this regime ensures that the gradient norm for high-surprisal tokens vanishes ($|\nabla| \to 0$). This behavior effectively imposes a bounded influence function on the estimator, preventing outliers from dominating the gradient updates and thereby immunizing the optimization process against residual artifacts. Remarkably, substituting $\beta = 0$ recovers the exact BLL-Loss formulation derived in Section 3.1:

$$\mathcal{L}_{\text{new}} = -\log \sigma\Big( -\log\big(1 - \pi_\theta(y_t \mid x)\big) \Big)$$
$$= \log\big(2 - \pi_\theta(y_t \mid x)\big). \tag{11}$$

This derivation establishes a profound **theoretical unification**: the geometric regularization (Perspective 1) and the conservative log-odds limit (Perspective 2) converge to the *identical* objective.

**Empirical Verification.** We validate this theoretical deduction in Figure 3. The results demonstrate a monotonic performance improvement as $\beta \to 0$, with the peak performance strictly located at the limit. This empirically confirms that robust alignment is best achieved by relying solely on Complement Suppression, effectively filtering out the instability induced by maximizing potentially erroneous targets. Consequently, we adopt the BLL configuration as our final objective.

### 3.3. Theoretical Analysis: Boundedness and Gradient Dynamics

**I. Elimination of Singularity (Functional Boundedness).** The standard $\mathcal{L}_{\text{CE}}$ is unbounded, exhibiting a singularity at $\pi_\theta(y_t \mid x) \to 0$ ($\lim_{\pi \to 0} -\log \pi = \infty$). This unbounded penalty forces the model to overfit outliers or noisy samples to reduce the diverging loss. In contrast, $\mathcal{L}_{\text{BLL}}$ is strictly bounded:

$$0 \le \mathcal{L}_{\text{BLL}} = \log(2 - \pi_\theta(y_t \mid x)) \le \log 2. \tag{12}$$

As $\pi_\theta(y_t \mid x) \to 0$, the loss saturates at $\log 2 \approx 0.69$. This saturation acts as an intrinsic *outlier rejection mechanism*, capping the penalty for low-probability tokens and preventing numerical instability during training.

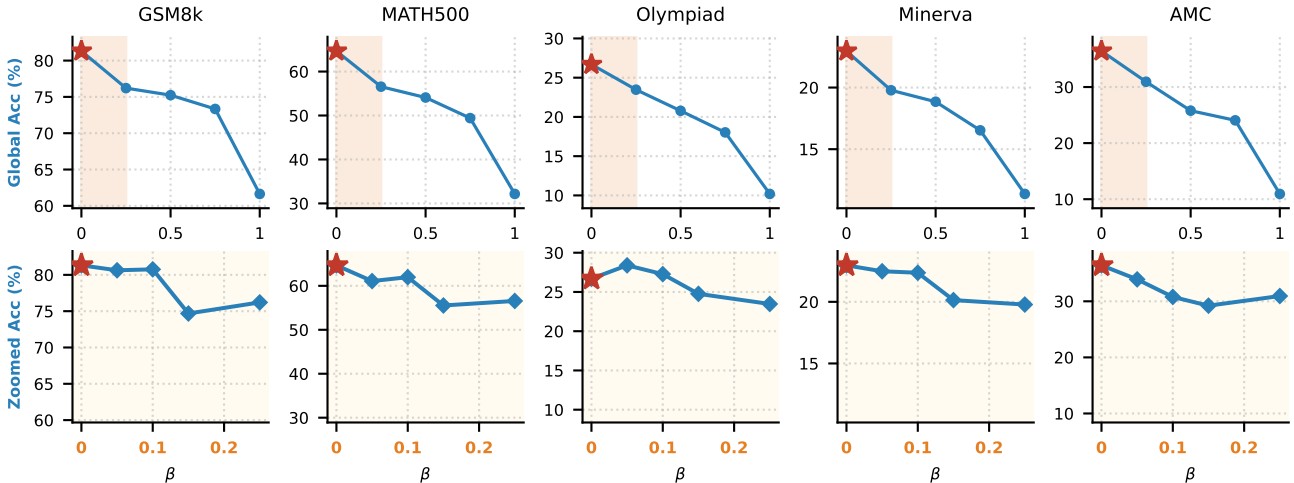

*Figure 3.* Sensitivity analysis of the hyperparameter $\beta$ across five mathematical reasoning benchmarks. The top row shows the overall performance trend over $\beta \in [0, 1]$. The bottom row presents a magnified view of the shaded region in the top row, corresponding to $\beta \in [0, 0.25]$, to highlight fine-grained behavior under small perturbations. The red star marks the baseline performance at $\beta = 0$.

**II. Soft-Screening Mechanism (Gradient Dynamics).** To understand how this boundedness affects optimization, we analyze the gradient with respect to the pre-softmax logits $z_t$. Applying the chain rule, we derive:

$$\nabla_{z_t} \mathcal{L}_{\text{BLL}} = \underbrace{\left( \frac{\pi_\theta(y_t \mid x)}{2 - \pi_\theta(y_t \mid x)} \right)}_{\text{Modulation } \lambda(\pi_\theta(y_t \mid x))} \cdot \underbrace{(\pi_\theta(y_t \mid x) - 1)}_{\nabla_{z_t} \mathcal{L}_{\text{CE}}}. \quad (13)$$

Eq. (13) reveals that BLL naturally re-weights the standard CE gradient by a modulation factor $\lambda(\pi_\theta(y_t \mid x))$. Comparing the asymptotic gradient norms as $\pi_\theta(y_t \mid x) \to 0$, we observe a sharp contrast:

$$\lim_{\pi_\theta(\cdot) \to 0} |\nabla_{z_t} \mathcal{L}_{\text{CE}}| = 1, \quad \text{whereas} \quad \lim_{\pi_\theta(\cdot) \to 0} |\nabla_{z_t} \mathcal{L}_{\text{BLL}}| = 0. \quad (14)$$

While CE applies a constant, maximal force to correct mis-classified tokens (even if they are noise), BLL implements a soft-screening strategy: the gradient vanishes as the target probability approaches zero. This ensures that the model updates are driven by learnable, high-confidence patterns rather than being disrupted by intractable hard samples or stylistic artifacts. More analysis can be found in Appendix B.1.

# 4. Experiments

## 4.1. Implementation Details

We conduct supervised fine-tuning on a diverse set of pre-trained and instruction-tuned baselines, specifically Qwen2.5-Math (1.5B/7B) (Yang et al., 2024), Qwen3-Base (1.7B/4B/8B) (Yang et al., 2025), and Llama-3

(1B/8B) (Grattafiori et al., 2024). For the training corpus, we utilize the NuminaMath-CoT dataset[2], which aggregates challenging mathematical reasoning problems ranging from Chinese high school exercises to US and international Olympiad competitions. To balance computational efficiency with data diversity, we randomly subsample 100k instances from the dataset for training. All models are fine-tuned for a single epoch with a learning rate of $1 \times 10^{-5}$ and a global batch size of 256. During inference, we employ a temperature of 1.0 and a maximum generation length of 8,192 tokens. We additionally compare against two optimization-oriented baselines, NEFT (Jain et al., 2024) and GEM (Li et al., 2025b). To mitigate generation stochasticity and ensure statistical reliability, we evaluate performance on a comprehensive suite of benchmarks (e.g., GSM8K (Cobbe et al., 2021), Math500 (Hendrycks et al., 2021), Gaokao2023 (Zhang et al., 2023)). We report the mean accuracy averaged over 16 independent evaluation runs. Detailed experimental settings are provided in the Appendix C.

## 4.2. Main Results

Table 1 presents the main results across a diverse set of mathematical reasoning benchmarks. Across all model architectures and scales, BLL-Loss consistently outperforms standard SFT and its variants, with performance gains that persist as model size increases. In several settings, BLL-Loss improves average accuracy by double-digit margins relative to both the Base model and standard SFT. Beyond overall improvements, a qualitative difference emerges between BLL-Loss and cross-entropy-based fine-tuning meth-

---

[2]https://huggingface.co/datasets/AI-MO/NuminaMath-CoT

*Table 1.* Main Results on Mathematical Reasoning Benchmarks. We categorize datasets into General Reasoning, Academic & Scientific, and Competition Math domains. The Comp. Avg column denotes the average accuracy on competition datasets. Bold indicates the best performance within each model scale. Llama3 results are reported in Appendix D.2.

| Models | Methods | General Math | | Academic & Sci. | | Competition Math | | | | Overall | |
|---|---|---|---|---|---|---|---|---|---|---|---|
| | | GSM8K | Math500 | Gaokao | Minerva | Olympiad | AMC | AIME | *Avg.* | Avg. | Δ |
| Qwen2.5-Math-1.5B | Base | 31.06 | 31.90 | 24.26 | 8.01 | 15.97 | 17.81 | 2.49 | *12.09* | 18.79 | - |
| | GEM | 54.50 | 13.20 | 20.80 | 3.70 | 3.60 | 2.50 | 0.00 | *2.03* | 14.04 | -4.75 |
| | SFT | 72.61 | 43.26 | 39.11 | 11.86 | 13.82 | 17.03 | 0.83 | *10.56* | 28.36 | +9.57 |
| | NEFT | 71.73 | 45.80 | 41.96 | 15.49 | 16.63 | 21.56 | 1.66 | *13.28* | 30.69 | +11.90 |
| | **BLL** | **81.31** | **64.66** | **53.28** | **22.96** | **26.69** | **36.41** | **5.62** | *22.91* | **41.56** | **+22.77** |
| Qwen2.5-Math-7B | Base | 41.06 | 39.43 | 25.10 | 12.83 | 17.37 | 28.75 | 9.18 | *18.43* | 26.05 | - |
| | GEM | 85.13 | 43.04 | 37.14 | 8.41 | 6.43 | 17.19 | 1.66 | *8.43* | 28.43 | +2.38 |
| | SFT | 83.44 | 53.50 | 44.94 | 18.26 | 19.94 | 28.44 | 3.12 | *17.17* | 35.95 | +9.90 |
| | NEFT | 82.55 | 52.26 | 45.80 | 18.96 | 19.86 | 25.94 | 1.24 | *15.68* | 35.23 | +9.18 |
| | **BLL** | **87.96** | **69.10** | **56.45** | **27.98** | **32.23** | **41.72** | **6.70** | *26.88* | **46.46** | **+20.41** |
| Qwen3-1.7B-Base | Base | 47.76 | 33.72 | 28.53 | 10.05 | 12.04 | 14.69 | 0.62 | *9.12* | 21.06 | - |
| | SFT | 63.57 | 35.65 | 32.28 | 8.16 | 8.52 | 11.25 | 0.41 | *6.73* | 22.83 | +1.77 |
| | NEFT | 64.80 | 35.66 | 33.13 | 8.05 | 9.10 | 13.91 | 1.24 | *8.08* | 23.70 | +2.64 |
| | **BLL** | **74.29** | **50.75** | **43.53** | **15.46** | **16.19** | **25.47** | **0.21** | *13.96* | **32.27** | **+11.21** |
| Qwen3-4B-Base | Base | 40.04 | 36.01 | 32.27 | 11.61 | 18.53 | 21.72 | 5.41 | *15.22* | 23.66 | - |
| | SFT | 80.69 | 48.96 | 42.27 | 15.78 | 17.17 | 22.50 | 1.24 | *13.64* | 32.66 | +9.00 |
| | NEFT | 80.30 | 49.85 | 42.43 | 16.16 | 17.57 | 24.84 | 1.44 | *14.62* | 33.23 | +9.57 |
| | **BLL** | **88.98** | **64.96** | **56.86** | **22.81** | **29.70** | **40.47** | **7.29** | *25.82* | **44.44** | **+20.78** |
| Qwen3-8B-Base | Base | 74.81 | 57.63 | 47.62 | 19.67 | 27.40 | 35.47 | 8.55 | *23.81* | 38.74 | - |
| | SFT | 84.40 | 52.93 | 44.95 | 18.11 | 19.38 | 25.00 | 1.88 | *15.42* | 35.24 | -3.50 |
| | NEFT | 83.29 | 52.78 | 45.46 | 17.33 | 18.91 | 22.97 | 3.74 | *15.21* | 34.93 | -3.81 |
| | **BLL** | **91.61** | **67.50** | **59.09** | **21.10** | **33.46** | **40.47** | **5.82** | *26.58* | **45.58** | **+6.84** |

ods. While SFT and NEFT achieve moderate gains on relatively easier benchmarks, their performance deteriorates on more challenging reasoning tasks, and in some cases even falls below that of the Base model. This contrast is most pronounced on competition-level mathematics benchmarks. As reflected in the *Competition Math* averages, BLL-Loss consistently improves performance across all scales, whereas standard SFT often exhibits performance degradation. These results are consistent with our analysis that when fine-tuning relies heavily on token-level supervision tied to realization-specific or stylistic features, optimization pressure is misallocated, ultimately harming reasoning generalization.

### 4.3. Ablation Study

Inherently, our BLL-Loss operates by suppressing gradient magnitudes, limiting overfitting to low-probability tokens. However, the observed improvements might merely stem from restricted update amplitudes rather than the objective landscape itself. To investigate this, we rigorously compare BLL-Loss against standard SFT baselines under explicit optimization constraints, such as reduced learning rates and gradient clipping. Table 2 reveals a clear distinction between these approaches. We find that standard SFT is highly sensitive to step-size reduction; a lower learning rate ($1 \times 10^{-6}$) leads to catastrophic performance degradation (dropping

from 31.72% to 23.43%). While gradient clipping provides slight stabilization, it remains significantly inferior to BLL-Loss. These findings confirm that BLL-Loss offers a fundamental advantage: it selectively modulates the optimization landscape based on confidence, achieving a robustness that simple manual gradient constraints fail to emulate.

### 4.4. Data Scaling Efficiency

Beyond optimization dynamics, an effective objective should exhibit robustness across different data regimes. Fig-

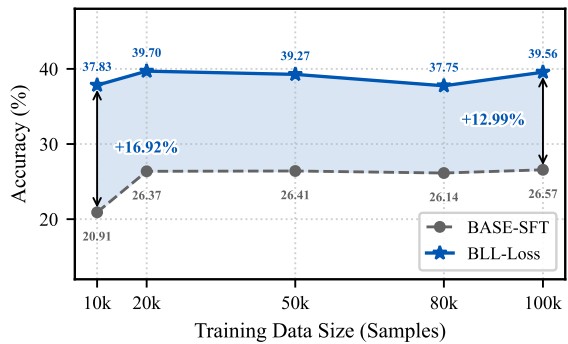

*Figure 4.* Performance of BLL-Loss and SFT across varying fractions of the training data (10k-100k). BLL outperforms the SFT at all data scales, with a stable performance gap as the training set size increases. Details can be found in Appendix C.4.

*Table 2.* Ablation & Scalability Analysis. Panel A details the optimization ablation on Qwen2.5-Math-1.5B. Panel B reports the same ablation study on the larger 7B model to verify scalability. BLL consistent outperforms SFT and its variants across all scales.

| Method | GSM8K | Math500 | Minerva | Olymp. | AMC | Avg. |
|---|---|---|---|---|---|---|
| **Panel A: Qwen2.5-Math-1.5B** | | | | | | |
| **Standard SFT** | 72.61 | 43.26 | 11.86 | 13.82 | 17.03 | 31.72 |
| *SFT Variants (Optimization Controls)* | | | | | | |
| Lower LR (1e−6) | 49.59 | 32.71 | 9.32 | 10.83 | 14.69 | 23.43 |
| Grad Clip (0.3) | 70.43 | 42.89 | 15.66 | 14.90 | 16.56 | 32.09 |
| Grad Clip (0.5) | 70.58 | 42.86 | 16.48 | 15.25 | 15.47 | 32.13 |
| More Epoch (2) | 73.58 | 45.16 | 15.39 | 16.31 | 19.21 | 33.93 |
| **BLL (Ours)** | **81.31** | **64.66** | **22.96** | **26.69** | **36.41** | **46.41** |
| **Panel B: Qwen2.5-Math-7B** | | | | | | |
| **Standard SFT** | 83.44 | 53.50 | 18.26 | 19.94 | 28.44 | 40.72 |
| *SFT Variants (Optimization Controls)* | | | | | | |
| Lower LR (1e−6) | 66.72 | 46.97 | 14.95 | 16.86 | 25.47 | 34.19 |
| Grad Clip (0.3) | 83.67 | 53.06 | 19.60 | 20.43 | 28.28 | 40.81 |
| Grad Clip (0.5) | 83.41 | 53.54 | 18.73 | 20.28 | 24.53 | 40.10 |
| More Epoch (2) | 86.54 | 58.19 | 20.79 | 22.94 | 29.53 | 43.60 |
| **BLL (Ours)** | **87.96** | **69.10** | **27.98** | **32.23** | **41.72** | **51.80** |

*Table 3.* Holistic Generalization Capabilities. Detailed breakdown of performance across different model scales and domains. We report accuracy on Code Generation (HumanEval, MBPP) and General Reasoning (MMLU, BBH, Commonsense). BLL achieves consistent improvements across all valid metrics.

| Model | Method | Code Generation | | | General Reasoning | | | |
|---|---|---|---|---|---|---|---|---|
| | | HE | MBPP | *Avg.* | MMLU | BBH | CS | *Avg.* |
| Qwen3-1.7B-Base | Base | 16.46 | 23.80 | 20.13 | 52.10 | 13.71 | 65.36 | 43.72 |
| | SFT | 44.51 | 25.20 | 34.86 | 53.50 | 33.78 | 65.93 | 51.07 |
| | **BLL** | **60.98** | **48.40** | **54.69** | **62.60** | **49.18** | **75.84** | **62.54** |
| Qwen3-4B-Base | Base | 46.95 | 44.40 | 45.68 | 65.23 | 11.76 | 74.77 | 50.59 |
| | SFT | 70.12 | 38.40 | 54.26 | 67.46 | 51.49 | 77.56 | 65.50 |
| | **BLL** | **73.80** | **61.80** | **67.80** | **71.76** | **65.85** | **81.49** | **73.03** |

ure 4 compares the performance of BLL-Loss and standard SFT as the training set size increases from 10k to 100k samples. Across all data scales, BLL-Loss consistently achieves higher accuracy than standard SFT, indicating improved sample efficiency. Notably, BLL-Loss trained on only 10k samples (approximately 10% of the full dataset) already surpasses the performance of standard SFT trained on the entire 100k dataset. This result suggests that BLL-Loss more effectively leverages limited supervision by prioritizing reasoning-relevant training signals over realization-level artifacts, leading to improved data efficiency.

### 4.5. Code & General Reasoning

While our primary analysis focuses on mathematical reasoning, we further examine whether the benefits of BLL-Loss extend to other reasoning-related domains. To this end, we evaluate BLL-Loss on code generation, trained on oss_instruct (Wei et al., 2024), and on general reasoning, trained on alpaca-gpt4 (Peng et al., 2023). As reported in Table 3, BLL-Loss consistently outperforms standard SFT across these settings, suggesting that its advantages are not confined to mathematical problem solving. Notably, prior to fine-tuning, Qwen3-1.7B-Base slightly outperforms its 4B counterpart on BBH . We attribute this anomaly to the prompt sensitivity and output format collapse in unaligned base models, rather than intrinsic reasoning gaps. Once properly aligned, the expected scaling law is restored, with the 4B model significantly surpassing the 1.7B model. Furthermore, BLL-Loss substantially improves the Qwen3-4B model, increasing MBPP accuracy to 61.8% (+23.40% over SFT) and yielding a +14.36% gain on BBH. These findings confirm that selectively suppressing low-confidence tokens is a fundamentally domain-agnostic principle for reasoning.

## 5. Related Work

**Supervised Fine-Tuning.** SFT is typically formulated as maximum likelihood estimation with a token-level cross-entropy objective and remains the dominant approach for adapting pre-trained language models to downstream tasks (Ouyang et al., 2022; Grattafiori et al., 2024; Yang et al., 2025). Prior work has improved SFT through regularization or optimization-level techniques, including label smoothing (Szegedy et al., 2016), representation noise injection such as NEFTune (Jain et al., 2024), conservative hyperparameter choices (Lin et al., 2025), and loss designs encouraging output diversity such as GEM (Li et al., 2025b). While these methods operate within the standard SFT framework and apply uniform token-level supervision without explicitly distinguishing token roles, recent concurrent works have explored objective modifications. For instance, Li et al. (2025a) empirically find that replacing the standard cross-entropy loss with probability-based objectives significantly improves reasoning performance. Similarly, Wu et al. (2025) re-examine the implicit reward in SFT through a reinforcement learning perspective to propose a rescaled loss function. Unlike these approaches, our BLL-Loss is rigorously derived from two independent perspectives, offering a principled alternative that opens up broader avenues for future objective modifications.

**Fine-Tuning for Reasoning.** Prior work indicates that supervised fine-tuning mainly elicits or shapes knowledge acquired during pre-training, rather than injecting new knowledge (Zhou et al., 2023a). However, when this superficial alignment perspective is applied to complex reasoning tasks, standard supervised fine-tuning often exhibits clear optimization limitations. Empirical analyses indicate that cross-entropy-based SFT objectives tend to encourage models to mechanically fit training trajectories, rather than learning function mappings that generalize beyond the fine-tuning distribution (Lobo et al., 2025; Jin et al., 2025; Chu et al., 2025). Recent studies on advanced reasoning models (e.g., R1-style models) further highlight these limitations (Chen et al., 2025). However, prior analyses of these limitations

remain largely macroscopic. We advance this understanding by identifying the mechanistic root cause, specifically the optimization interference induced by fitting stylistic tokens, and accordingly provide a direct, theoretically grounded remedy.

## 6. Conclusion

In this work, we analyze SFT for reasoning from a token-level perspective and show that standard cross-entropy implicitly treats all tokens as equally reliable, which is misaligned with the inherent variability of valid reasoning realizations. We find that low-probability tokens are often realization-level artifacts yet induce disproportionate optimization pressure under standard SFT. To address this issue, we propose BLL-Loss, a loss-level objective that introduces intrinsic boundedness to suppress excessive optimization pressure induced by low-probability tokens, while preserving stable discriminative constraints. Extensive experiments across model scales and data regimes demonstrate that BLL-Loss consistently improves reasoning performance.

## 7. Limitations and Future Work

While we empirically validate BLL-Loss across mathematical, code, and language reasoning, its applicability to multimodal domains, such as visual reasoning, remains an open question. Furthermore, our approach fundamentally relies on model confidence to modulate token-level supervision, effectively eliciting reasoning capabilities acquired during pre-training. Consequently, under severe distribution shifts where the target domain deviates significantly from the pretrained knowledge, BLL-Loss may underperform standard cross-entropy due to the lack of reliable priors. Future work could explore adaptive mechanisms that dynamically interpolate between standard CE and BLL-Loss based on the model's intrinsic domain familiarity.

## Acknowledgments

This work was supported in part by the National Key R&D Program of China under Grant 2023YFF0905503, National Natural Science Foundation of China under Grants No.62472188, The Education University of Hong Kong Start-up Funding Grant R4400 and EdUHK Seed Funding Grant R4539.

## Impact Statement

This paper presents work whose goal is to advance the field of Machine Learning. There are many potential societal consequences of our work, none which we feel must be specifically highlighted here.

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

## A. Derivation of Log-Odds Decomposition

We provide a rigorous derivation of the isomorphic mapping between standard Cross-Entropy and the Log-Odds objective. Let $\mathcal{V}$ denote the vocabulary set. For a given context $x$, the model outputs a probability distribution $\pi_\theta(\cdot \mid x)$ over $\mathcal{V}$, satisfying the simplex constraint:

$$\sum_{y \in \mathcal{V}} \pi_\theta(y \mid x) = 1. \tag{15}$$

We partition the vocabulary into the target token $y_t$ and its complement set $\mathcal{V} \setminus \{y_t\}$. Consequently, the normalization condition can be expanded as the sum of the target probability and the cumulative mass of non-target tokens:

$$\pi_\theta(y_t \mid x) + \sum_{y' \in \mathcal{V} \setminus \{y_t\}} \pi_\theta(y' \mid x) = 1. \tag{16}$$

Using this expansion, we rewrite the target probability $\pi_\theta(y_t \mid x)$ as a self-normalized ratio:

$$\pi_\theta(y_t \mid x) = \frac{\pi_\theta(y_t \mid x)}{\pi_\theta(y_t \mid x) + \sum_{y' \neq y_t} \pi_\theta(y' \mid x)}. \tag{17}$$

To align this with the logistic sigmoid function $\sigma(z) = (1 + e^{-z})^{-1}$, we divide both the numerator and the denominator by the target probability $\pi_\theta(y_t \mid x)$:

$$\pi_\theta(y_t \mid x) = \frac{1}{1 + \frac{\sum_{y' \neq y_t} \pi_\theta(y' \mid x)}{\pi_\theta(y_t \mid x)}}. \tag{18}$$

Next, we transform the ratio in the denominator into the logarithmic domain using the identity $a/b = \exp(\log a - \log b)$:

$$\begin{aligned} \frac{\sum_{y' \neq y_t} \pi_\theta(y' \mid x)}{\pi_\theta(y_t \mid x)} &= \exp\left( \log\left( \sum_{y' \neq y_t} \pi_\theta(y' \mid x) \right) - \log \pi_\theta(y_t \mid x) \right) \\ &= \exp\left( - \left[ \log \pi_\theta(y_t \mid x) - \log\left( \sum_{y' \neq y_t} \pi_\theta(y' \mid x) \right) \right] \right). \end{aligned} \tag{19}$$

Substituting this back into the fraction yields the sigmoid form:

$$\pi_\theta(y_t \mid x) = \sigma\left( \log \pi_\theta(y_t \mid x) - \log\left( \sum_{y' \neq y_t} \pi_\theta(y' \mid x) \right) \right). \tag{20}$$

Finally, substituting this expression into the standard Cross-Entropy loss $\mathcal{L}_{\text{CE}} = -\log \pi_\theta(y_t \mid x)$, and recognizing that the cumulative sum of non-target probabilities is simply the complement probability $1 - \pi_\theta(y_t \mid x)$, we arrive at the proposed decomposition:

$$\begin{aligned} \mathcal{L}_{\text{CE}} &= -\log \sigma\left( \underbrace{\log \pi_\theta(y_t \mid x)}_{\text{Target Promotion}} - \underbrace{\log\left( \sum_{y' \neq y_t} \pi_\theta(y' \mid x) \right)}_{\text{Complement Suppression}} \right) \\ &= -\log \sigma\left( \log \pi_\theta(y_t \mid x) - \log\left( 1 - \pi_\theta(y_t \mid x) \right) \right). \end{aligned} \tag{21}$$

## B. Theoretical Analysis and Detailed Derivations

### B.1. Gradient with respect to target logits

In this section, we provide the step-by-step derivation of the gradient of the proposed Bounded Log-Odds Loss ($\mathcal{L}_{\text{BLL}}$) with respect to the pre-softmax logits. We explicitly retain the full probability notation $\pi_\theta(y_t \mid x)$ to maintain rigorous notation consistency with the main text.

**Preliminaries: The Jacobian of Softmax.** Let $z \in \mathbb{R}^{|\mathcal{V}|}$ denote the vector of logits output by the model, and let $z_t$ be the logit corresponding to the target token $y_t$. The probability $\pi_\theta(y_t \mid x)$ is given by the Softmax function:

$$\pi_\theta(y_t \mid x) = \frac{\exp(z_t)}{\sum_{j \in \mathcal{V}} \exp(z_j)}. \tag{22}$$

A fundamental property of the Softmax function is its derivative with respect to the logit $z_t$:

$$\frac{\partial \pi_\theta(y_t \mid x)}{\partial z_t} = \pi_\theta(y_t \mid x)\Big(1 - \pi_\theta(y_t \mid x)\Big). \tag{23}$$

**Derivation.** The objective function for a single token is defined as:

$$\mathcal{L}_{\text{BLL}}(\theta) = \log\Big(2 - \pi_\theta(y_t \mid x)\Big). \tag{24}$$

To find the gradient with respect to the parameter $\theta$, we first compute the gradient with respect to the target logit $z_t$ using the Chain Rule:

$$\nabla_{z_t}\mathcal{L}_{\text{BLL}} = \frac{\partial \mathcal{L}_{\text{BLL}}}{\partial \pi_\theta(y_t \mid x)} \cdot \frac{\partial \pi_\theta(y_t \mid x)}{\partial z_t}. \tag{25}$$

**Step 1: Compute the derivative of the loss w.r.t. probability.**

$$\frac{\partial \mathcal{L}_{\text{BLL}}}{\partial \pi_\theta(y_t \mid x)} = \frac{\partial}{\partial \pi}\log(2 - \pi) = \frac{-1}{2 - \pi_\theta(y_t \mid x)}. \tag{26}$$

**Step 2: Combine with the Softmax Jacobian.** Substituting the terms back into the Chain Rule equation:

$$\begin{aligned}
\nabla_{z_t}\mathcal{L}_{\text{BLL}} &= \left(\frac{-1}{2 - \pi_\theta(y_t \mid x)}\right) \cdot \Big[\pi_\theta(y_t \mid x)\Big(1 - \pi_\theta(y_t \mid x)\Big)\Big] \\
&= -\frac{\pi_\theta(y_t \mid x)\Big(1 - \pi_\theta(y_t \mid x)\Big)}{2 - \pi_\theta(y_t \mid x)}.
\end{aligned} \tag{27}$$

**Step 3: Factorization and Interpretation.** We observe that the standard Cross-Entropy gradient w.r.t. logits is $\nabla_{z_t}\mathcal{L}_{\text{CE}} = \pi_\theta(y_t \mid x) - 1$. We can rearrange the BLL gradient to reveal its relationship with CE:

$$\begin{aligned}
\nabla_{z_t}\mathcal{L}_{\text{BLL}} &= \frac{\pi_\theta(y_t \mid x)}{2 - \pi_\theta(y_t \mid x)} \cdot \Big(\pi_\theta(y_t \mid x) - 1\Big) \\
&= \lambda(\pi_\theta(y_t \mid x)) \cdot \nabla_{z_t}\mathcal{L}_{\text{CE}},
\end{aligned} \tag{28}$$

where $\lambda(\pi_\theta(y_t \mid x)) = \frac{\pi_\theta(y_t|x)}{2 - \pi_\theta(y_t|x)}$ serves as the dynamic modulation factor. This completes the derivation.

**B.2. Gradient with respect to non-target logits.**

For completeness, we also derive the gradient of $\mathcal{L}_{\text{BLL}}$ with respect to any non-target logit $z_k$ where $k \neq y_t$.

First, recall the Softmax Jacobian for non-target logits:

$$\frac{\partial \pi_\theta(y_t \mid x)}{\partial z_k} = -\pi_\theta(y_t \mid x)\pi_\theta(k \mid x). \tag{29}$$

Applying the chain rule, we compute the gradient of the BLL loss w.r.t. $z_k$:

$$\nabla_{z_k}\mathcal{L}_{\text{BLL}} = \frac{\partial \mathcal{L}_{\text{BLL}}}{\partial \pi_\theta(y_t \mid x)} \cdot \frac{\partial \pi_\theta(y_t \mid x)}{\partial z_k} \tag{30}$$

$$= \left(\frac{-1}{2 - \pi_\theta(y_t \mid x)}\right) \cdot (-\pi_\theta(y_t \mid x)\pi_\theta(k \mid x)) \tag{31}$$

$$= \frac{\pi_\theta(y_t \mid x)\pi_\theta(k \mid x)}{2 - \pi_\theta(y_t \mid x)}. \tag{32}$$

Recognizing that the gradient of standard Cross-Entropy for non-target logits is $\nabla_{z_k}\mathcal{L}_{\text{CE}} = \pi_\theta(k \mid x)$, we can rewrite this as:

$$\nabla_{z_k}\mathcal{L}_{\text{BLL}} = \underbrace{\left(\frac{\pi_\theta(y_t \mid x)}{2 - \pi_\theta(y_t \mid x)}\right)}_{\lambda(\pi_\theta(y_t \mid x))} \cdot \underbrace{\pi_\theta(k \mid x)}_{\nabla_{z_k}\mathcal{L}_{\text{CE}}}. \tag{33}$$

This establishes a profound mathematical property of the BLL-Loss: the exact same dynamic modulation factor $\lambda(\pi_\theta(y_t \mid x))$ that scales the target logit gradient synchronously scales the gradients for all non-target logits. Consequently, as $\pi_\theta(y_t \mid x) \to 0$, the gradient norm for the entire logit vector vanishes, effectively preventing noisy samples from incorrectly suppressing valid alternative tokens.

### B.3. Variance Analysis via Importance Sampling

To compare supervised objectives with RL, we must express the gradients as an expectation over the model's current policy $\pi_\theta$. Recall the standard Policy Gradient theorem:

$$\nabla J(\theta) = \mathbb{E}_{y \sim \pi_\theta(\cdot \mid x)}\Big[ \underbrace{R(y)}_{\text{Reward Signal}} \cdot \nabla_\theta \log \pi_\theta(y \mid x)\Big]. \tag{34}$$

Our goal is to identify the Implicit Reward $R(y)$ derived from both SFT and BLL losses to understand their learning dynamics.

**Standard SFT: The Unbounded Reward Problem** The standard Cross-Entropy loss is defined as $\mathcal{L}_{\text{CE}} = -\log \pi_\theta(y^\star \mid x)$. Its gradient is deterministic with respect to the ground truth $y^\star$. To map this to the stochastic RL form in Eq. (34), we employ **Importance Sampling (IS)**:

$$\begin{aligned}\nabla\mathcal{L}_{\text{SFT}} &= -\nabla \log \pi_\theta(y^\star \mid x) \\ &= -\mathbb{E}_{y \sim \pi_\theta}\left[\frac{\mathbb{1}[y = y^\star]}{\pi_\theta(y \mid x)} \cdot \nabla \log \pi_\theta(y \mid x)\right].\end{aligned} \tag{35}$$

Comparing this to Eq. (34), we isolate the implicit reward for SFT:

$$R_{\text{SFT}}(y) = \frac{1}{\pi_\theta(y)}. \tag{36}$$

**Theoretical Insight:** This reveals a critical instability. As $\pi_\theta(y^\star) \to 0$ (i.e., when the model is wrong or the data is noisy), the implicit reward $R_{\text{SFT}} \to \infty$. SFT fundamentally assumes that "rare" samples are infinitely valuable, causing gradient explosions.

**BLL: Intrinsic Reward Clipping** We now apply the same transformation to the Bounded Log-Odds Loss. Recall the BLL gradient derived in Appendix B.1: $\nabla\mathcal{L}_{\text{BLL}} = -\frac{\pi_\theta}{2-\pi_\theta}\nabla \log \pi_\theta$.

Mapping this to the RL expectation form:

$$\begin{aligned}\nabla\mathcal{L}_{\text{BLL}} &= -\mathbb{E}_{y \sim \pi_\theta}\left[\frac{\mathbb{1}[y = y^\star]}{\pi_\theta(y \mid x)} \cdot \left(\frac{\pi_\theta(y \mid x)}{2 - \pi_\theta(y \mid x)}\nabla \log \pi_\theta(y \mid x)\right)\right] \\ &= -\mathbb{E}_{y \sim \pi_\theta}\left[\underbrace{\frac{\mathbb{1}[y = y^\star]}{2 - \pi_\theta(y \mid x)}}_{R_{\text{BLL}}(y)} \cdot \nabla \log \pi_\theta(y \mid x)\right].\end{aligned} \tag{37}$$

**The Cancellation Effect:** The $\pi_\theta$ term in the BLL gradient numerator perfectly cancels the $1/\pi_\theta$ IS weight. The resulting implicit reward is:

$$R_{\text{BLL}}(y) = \frac{1}{2 - \pi_\theta(y \mid x)}. \tag{38}$$

**Comparative Analysis: BLL as Clipped RL**    This derivation establishes a direct theoretical equivalence between BLL and Reward Clipping mechanisms in RL.

| Framework | Implicit Reward $R(y)$ | Range | Mechanism Description |
|---|---|---|---|
| Standard SFT | $1/\pi$ | $[1, \infty)$ | **Inverse Propensity Weighting** (High Variance) |
| **BLL (Ours)** | $\mathbf{1/(2 - \pi)}$ | $[\mathbf{0.5, 1}]$ | **Clipped Importance Sampling** (Bounded Variance) |

*Table 4.* Comparison of Implicit Reward Structures.

**Relationship to PPO Clipping.**    While both BLL and PPO aim to enforce a Trust Region, they operate on different components:

- **PPO (Policy Clipping):** Clips the ratio $r_t = \pi_{\text{new}}/\pi_{\text{old}}$ to limit the *update step size*.

- **BLL (Reward Clipping):** Inherently clips the *reward signal* $R(y)$. Even for extreme outliers ($\pi \to 0$), the reward is capped at $0.5$ (via $1/(2 - \pi)$). This prevents single data points from exerting unbounded influence, effectively acting as an **intrinsic, data-dependent clipper** that stabilizes the learning dynamics without external hyperparameters.

## C. Implementation Details

### C.1. Training Implementation

**Training Framework.** To ensure reproducibility and standardize the training pipeline, we implemented all experiments using the **LLaMA-Factory** framework (Zheng et al., 2024). This unified library allowed us to conduct consistent full-parameter fine-tuning across different model architectures.

**Optimization & Infrastructure.** Given the computational demands of full-parameter fine-tuning, we utilized DeepSpeed ZeRO-3 with CPU Offloading (Rasley et al., 2020) to optimize memory usage. This configuration enabled efficient training of 7B-parameter models on a cluster of 2×NVIDIA A100 (80GB) GPUs. We employed `bfloat16` precision to maintain numerical stability.

**Data & Hyperparameters. Data Composition and Hyperparameters.** We constructed our training setups using three distinct datasets to evaluate different capabilities. For mathematical reasoning, we utilized the NuminaMath-CoT dataset, performing randomly sampling to select a subset of 100,000 challenging instances. Additionally, to benchmark general instruction-following capabilities, we conducted separate training experiments using the full training splits of **Alpaca-GPT4** and **OSS-Instruct**. Consistent with the configuration file, all models were fine-tuned for 1 epoch with a maximum sequence length of 2,048 tokens, utilizing the AdamW optimizer with a learning rate of $1 \times 10^{-5}$ and a linear decay scheduler. Regarding batch configuration, we adopted an adaptive strategy based on dataset scale. For all primary experiments, including the 100k NuminaMath run and the full-scale Alpaca/OSS runs, we maintained a global batch size of **256** to ensure stable gradient estimation. However, for the data scaling ablation studies (Section 4.4) where training data was restricted to subsets smaller than 100,000 samples, we reduced the global batch size to **32**. This adjustment was critical to ensure adequate optimization steps within limited data regimes.

### C.2. Evaluation Details

To strictly align with the hierarchy of our main results, we classify the evaluation benchmarks into three distinct categories: **General Math**, **Academic & Scientific Reasoning**, and **Competition Math**. This taxonomy allows us to evaluate the model's performance spectrum from foundational arithmetic to elite-level problem solving.

**I. General Math.**    These datasets serve as the baseline for evaluating fundamental mathematical reasoning and generalization capabilities.

- **GSM8K** (Cobbe et al., 2021): A high-quality dataset of 8.5k grade school math word problems. It primarily tests multi-step arithmetic and basic logic, serving as a standard proxy for fundamental reasoning ability.

- **Math500** (Hendrycks et al., 2021): A comprehensive test set consisting of 500 representative problems subsampled from the challenging MATH dataset.

**II. Academic & Scientific Reasoning.**     This category evaluates the model's ability to handle standardized academic tests and broader scientific reasoning tasks.

- **Gaokao** (Zhang et al., 2023): Derived from the 2023 Chinese National College Entrance Examination. We utilize the English-translated version of the mathematics section.

- **Minerva Math** (Lewkowycz et al., 2022): This benchmark tests the model's ability to apply mathematical concepts to physics, chemistry, and other STEM-related scenarios, requiring both domain knowledge and calculation precision.

**III. Competition Math.**     These benchmarks representing the upper bound of current LLM reasoning, consisting of problems designed for math olympiad participants.

- **Olympiad**: (Often referred to as OlympiadBench). A rigorous dataset aggregated from international mathematics competitions (e.g., IMO, CMO). It features open-ended problems that require constructing long, complex proofs and derivations.

- **AMC (American Mathematics Competitions)**We utilize the 2023 AMC problem sets. These are multiple-choice examinations in high school mathematics designed to promote the development of problem-solving skills.

- **AIME (American Invitational Mathematics Examination)**Specifically the 2024 version. The AIME is an intermediate examination between the AMC and the USAMO. The problems are much more difficult than the AMC.

**IV. Code Generation.**     To assess whether mathematical fine-tuning preserves programming capabilities (avoiding catastrophic forgetting) or enables code-aided reasoning, we evaluate on two standard Python benchmarks.

- **HumanEval** (Chen et al., 2021): Consists of 164 hand-written programming problems. Each problem includes a function signature, docstring, and unit tests. It evaluates the model's ability to interpret natural language instructions and synthesize correct algorithmic logic.

- **MBPP (Mostly Basic Python Programming)** (Austin et al., 2021): A crowd-sourced dataset containing around 1,000 entry-level Python programming tasks. It covers programming fundamentals, standard library usage, and simple data processing, providing a baseline for coding proficiency.

**V. General Reasoning & Common Sense.**     To evaluate the model's versatility beyond strict mathematical domains, we employ a set of diverse reasoning benchmarks.

- **MMLU (Massive Multitask Language Understanding)** (Hendrycks et al., 2020): A comprehensive benchmark covering 57 distinct tasks including elementary mathematics, US history, computer science, and law.

- **BBH (Big Bench Hard)** (Suzgun et al., 2023): A subset of 23 challenging tasks from the BIG-Bench suite where standard language models previously struggled. It requires multi-step logical deduction and tests the model's ability to follow complex instructions.

- **CommonsenseQA** (Talmor et al., 2019): A multiple-choice question answering dataset that requires distinct types of commonsense knowledge (e.g., spatial, social, causal) to predict the correct answer. This tests whether the model maintains grounding in real-world concepts despite being heavily tuned on abstract mathematical tokens.

To ensure reproducibility and align with community benchmarks, we implemented our evaluation pipeline using established open-source frameworks tailored to specific domains.

**Mathematical Reasoning Pipeline.** For all mathematical benchmarks (including GSM8K, Math500, Gaokao, Minerva, Olympiad, AMC, and AIME), we utilized the **official evaluation codebase** provided by Qwen2.5-Math.

- **Prompting & Extraction:** We strictly adhered to the official prompt templates (Zero-shot CoT) and answer extraction logic. This pipeline includes robust parsers specifically designed to handle complex mathematical notations (e.g., LaTeX formulas) and accurately extract final answers from the `\boxed{}` format or standard delimiters.

**Code & General Reasoning Pipeline.** For code generation (HumanEval, MBPP) and general reasoning tasks (MMLU, BBH, CommonsenseQA), we employed the **OpenCompass** evaluation framework (Contributors, 2023).

- **Implementation:** OpenCompass standardizes the evaluation process by applying its **built-in default settings** for task-specific prompt construction (e.g., canonical prompts for code, standard few-shot protocols for MMLU) and executing code solutions in a secure, sandboxed environment.

**Inference Configuration.** While utilizing these standard frameworks for prompt construction and scoring, we customized the inference parameters to ensure statistical reliability. Given the non-deterministic nature of generation, reporting a single run can be susceptible to variance. Therefore, we configured the frameworks to use a sampling temperature of $T = 1.0$ and conducted **16 independent generation runs** for each problem. We report the mean accuracy (Pass@1) averaged across these runs to minimize the impact of generation stochasticity.

### C.3. Baseline Details

We compare our method with the following baselines:

- **SFT**: This serves as our primary baseline. We perform full-parameter supervised fine-tuning on the pre-trained backbone using the standard autoregressive objective. Specifically, the model is trained to minimize the negative log-likelihood (Cross-Entropy Loss) of the target tokens given the input prompt.

- **NEFT** (Jain et al., 2024): A regularization method that adds random noise to the embedding vectors during fine-tuning to improve robustness and prevent overfitting. Following the original paper, we set the noise scale parameter $\alpha = 5$.

- **GEM-Loss** (Li et al., 2025b) : A recently proposed training objective designed to preserve output diversity during SFT. Unlike standard cross-entropy which forces strict distribution matching, GEM formulates SFT as a reverse-KL divergence minimization problem with an entropy regularizer. This approach prevents over-memorization and improves test-time scaling capabilities.

### C.4. Data Scaling Details

To study the effect of training data scale, we subsample 10k, 20k, 50k, and 80k examples from the 100k training set to fine-tune Qwen2.5-Math-1.5B. All resulting models are evaluated on a suite of mathematical reasoning benchmarks, including GSM8K, Math500, Minerva, Olympiad, AMC, and AIME.

## D. Additional Experimental Results

### D.1. More analysis of Tokens Probability

To better understand the nature of low-probability tokens in reasoning demonstrations, Figure 5 visualizes the semantic evolution of tokens across predicted probability intervals. The visualization reveals that the model's confidence is not distributed randomly, but is strictly aligned with the logical density of the content. Specifically, the high-probability regime (Interval $[0.75, 1.0]$) converges decisively on stable mathematical primitives and operators (e.g., "frac", "sqrt", "x"), whereas the low-probability tail (Interval $[0, 0.25)$) is saturated with variable narrative scaffolding (e.g., "Given", "Thus"). This semantic stratification provides compelling evidence that the pre-trained model already possesses robust inherent capabilities to recognize invariant reasoning structures. It confirms that the model's intrinsic prior $\pi_{\theta_0}$ effectively distinguishes between core logical execution (signal) and stylistic surface forms (noise), thereby justifying the use of model confidence as a proxy for token quality. Note that to prevent visual clutter from the dense use of digits and punctuation in mathematics, this visualization exclusively aggregates alphabetical tokens and LaTeX commands.

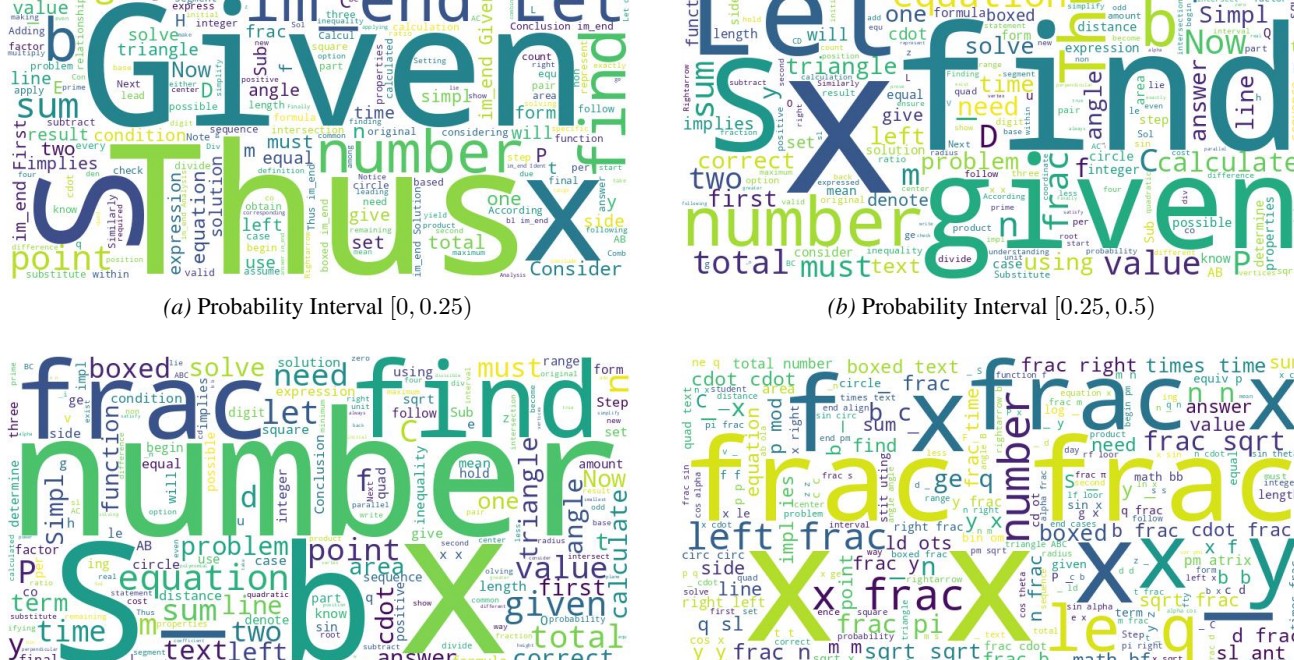

*(a)* Probability Interval $[0, 0.25)$

*(b)* Probability Interval $[0.25, 0.5)$

*(c)* Probability Interval $[0.5, 0.75)$

*(d)* Probability Interval $[0.75, 1.0]$

*Figure 5.* The figures display the most frequent tokens in the NuminaMath dataset, grouped by the model's predicted probability $p_\theta(y|x)$ into four intervals: (a) $0 \leq p < 0.25$, (b) $0.25 \leq p < 0.5$, (c) $0.5 \leq p < 0.75$, and (d) $0.75 \leq p \leq 1.0$.

### D.2. Generalization on Llama3 Family Models

To verify the architectural universality of our proposed method, we extended our evaluation to the Llama-3 family, specifically **Llama-3.2-1B** and **Llama-3.1-8B**. Unlike the Qwen-Math series which are initialized from math-specialized checkpoints, these Llama models utilize general-purpose base models.

The results are presented in Table 5.

- **On Llama-3.2-1B:** Due to the limited model capacity and lack of mathematical pre-training, the baseline performance is near zero. Standard SFT yields marginal improvements (Avg. 2.41%). However, **BLL-Loss** significantly amplifies this gain, achieving an average accuracy of **4.38%**, nearly doubling the performance of SFT.

- **On Llama-3.1-8B:** Standard SFT provides a massive boost over the base model (raising GSM8K from 2.07% to 58.38%). Incorporating BLL-Loss further enhances this strong baseline, particularly on challenging competition benchmarks (e.g., **+3.75%** on AMC), demonstrating that our method effectively mitigates overfitting even when learning from scratch on general base models.

These findings confirm that BLL-Loss is robust across different model architectures and initialization states (specialized vs. general).

### D.3. Comparison with Concurrent Works

To further contextualize our approach, we compare BLL-Loss against two concurrent token-level objectives: the probability-based objective (denoted as $-p$) (Li et al., 2025a) and Dynamic Fine-Tuning (DFT) (Wu et al., 2025). We train Qwen3-1.7B and Qwen3-4B on a randomly sampled 10k subset of NuminaMath-CoT, strictly preserving the hyperparameter configurations from our main experiments. As shown in Table 6, BLL-Loss consistently achieves the highest overall accuracy and demonstrates particularly strong performance on the challenging Competition Math benchmarks.

*Table 5.* **Additional Results on Llama-3 Family.** We compare BLL-Loss against standard SFT on Llama-3.2-1B and Llama-3.1-8B. *Note:* Since these are general-purpose base models (unlike Qwen-Math), the zero-shot performance on competition math is naturally lower. **BLL** consistently outperforms SFT across both scales.

| Models | Methods | General Math | | Academic & Sci. | | Competition Math | | | | Overall | |
| --- | --- | --- | --- | --- | --- | --- | --- | --- | --- | --- | --- |
| | | GSM8K | Math500 | Gaokao | Minerva | Olympiad | AMC | AIME | *Avg.* | Avg. | Δ |
| | Base | 1.22 | 1.78 | 3.38 | 0.84 | 0.98 | 1.41 | 0.00 | *0.80* | 1.37 | - |
| Llama-3.2-1B | SFT | 4.64 | 2.95 | 5.41 | 0.79 | 1.20 | 1.88 | 0.00 | *1.03* | 2.41 | +1.04 |
| | **BLL** | **10.41** | **5.51** | **8.27** | **2.01** | **1.79** | **2.66** | 0.00 | *1.48* | **4.38** | **+3.01** |
| | Base | 2.07 | 2.16 | 3.64 | 1.61 | 0.73 | 1.40 | 0.00 | *0.71* | 1.66 | - |
| Llama-3.1-8B | SFT | 58.38 | **20.81** | 23.14 | 4.94 | 4.65 | 7.19 | 0.00 | *3.95* | 17.02 | +15.36 |
| | **BLL** | **60.52** | 20.70 | **27.16** | **5.96** | **5.33** | **10.94** | 0.00 | *5.42* | **18.66** | **+17.00** |

*Table 6.* **Comparison with Concurrent Token-Level Objectives.** Evaluation of BLL-Loss against the probability-based objective ($-p$) (Li et al., 2025a) and DFT (Wu et al., 2025). Models are fine-tuned on a 10k subset of NuminaMath-CoT using Qwen3 base architectures.

| Models | Methods | General Math | | Academic & Sci. | | Competition Math | | | | Overall |
| --- | --- | --- | --- | --- | --- | --- | --- | --- | --- | --- |
| | | GSM8K | Math500 | Gaokao | Minerva | Olympiad | AMC | AIME | *Avg.* | Avg. |
| | -P | 76.76 | 54.88 | 47.12 | 17.83 | 20.32 | 29.06 | 0.83 | *16.74* | 35.26 |
| Qwen3-1.7B-Base | DFT | 76.43 | 55.90 | 47.74 | 16.71 | 21.19 | 29.84 | 1.66 | *17.56* | 35.64 |
| | **BLL** | 75.15 | 55.53 | 47.56 | 17.94 | 21.86 | 26.88 | 4.99 | ***17.91*** | **35.70** |
| | -P | 88.63 | 64.16 | 54.77 | 22.57 | 29.04 | 31.41 | 8.33 | *22.93* | 42.70 |
| Qwen3-4B-Base | DFT | 85.71 | 63.11 | 55.34 | 23.59 | 30.26 | 31.56 | 6.25 | *21.69* | 42.26 |
| | **BLL** | 88.64 | 64.09 | 55.72 | 18.05 | 30.19 | 35.31 | 9.17 | ***24.89*** | **43.02** |

### D.4. Failure Cases: Knowledge Injection

To further delineate the applicable boundaries of our method, we conducted additional validation on a model-weak task, specifically utilizing the challenging 40K induced dataset reported in Li et al. (2025a). On this highly out-of-distribution dataset, our BLL-Loss achieved a score of 0. This extreme performance degradation is entirely consistent with the findings in concurrent literature. The underlying reason for this failure is structurally inherent to our method. BLL-Loss relies on the model's pre-trained priors to distinguish core signal from stylistic noise. In model-weak scenarios, the model lacks the prerequisite knowledge and thus assigns inherently low probabilities to the correct, novel reasoning tokens. Consequently, BLL-Loss mistakenly interprets these essential learning signals as low-confidence noise and overly attenuates their gradients. This prevents the model from acquiring the necessary new capabilities. This failure case starkly confirms our hypothesis: BLL-Loss is uniquely suited for eliciting and aligning existing reasoning priors, but it is fundamentally unsuitable for injecting entirely novel knowledge into models.

### D.5. Full MMLU Breakdown

Due to space constraints in the main text, we provide the comprehensive breakdown of the MMLU evaluation in Table 7. We evaluate both Qwen3-1.7B and Qwen3-4B models across all 57 subjects, covering domains such as STEM, humanities, and social sciences. The results demonstrate that our proposed method consistently outperforms the SFT baseline across diverse tasks. Notably, we observe significant improvements in reasoning-intensive subjects (e.g., College Mathematics, Formal Logic) as well as knowledge-heavy subjects (e.g., Clinical Knowledge). This confirms that our approach enhances the model's general capabilities without compromising performance in specific domains. The reported scores represent the accuracy (%) .

### D.6. Full BBH Breakdown

Table 8 details the performance on the Big Bench Hard (BBH) benchmark, which serves as a rigorous testbed for complex reasoning capabilities beyond factual knowledge. Our method yields consistent and substantial gains over the SFT baseline

*Table 7.* Full MMLU breakdown. 'H.S.' denotes High School, 'Coll.' denotes College. Best performance in each model group is bolded.

| Subject | Qwen3-1.7B | | Qwen3-4B | | Subject | Qwen3-1.7B | | Qwen3-4B | |
|---|---|---|---|---|---|---|---|---|---|
| | SFT | Ours | SFT | Ours | | SFT | Ours | SFT | Ours |
| Abstract Algebra | 28.00 | **29.00** | 45.00 | **55.00** | H.S. Statistics | 45.83 | **58.80** | 67.59 | **75.00** |
| Anatomy | 53.33 | **57.04** | **65.19** | 64.44 | H.S. US History | 61.27 | **72.06** | 75.00 | **80.39** |
| Astronomy | 55.92 | **68.42** | 76.97 | **79.61** | H.S. World History | 65.40 | **73.84** | 75.95 | **81.01** |
| Business Ethics | 51.00 | **65.00** | 76.00 | **77.00** | Human Aging | 54.71 | **64.13** | 62.78 | **70.85** |
| Clinical Knowledge | 60.00 | **67.92** | 69.06 | **73.96** | Human Sexuality | 56.49 | **71.76** | 68.70 | **74.05** |
| Coll. Biology | 57.64 | **71.53** | 73.61 | **79.17** | International Law | 61.98 | **71.07** | 72.73 | **80.17** |
| Coll. CS | 50.00 | **58.00** | 54.00 | **70.00** | Jurisprudence | 56.48 | **71.30** | 71.30 | 72.22 |
| Coll. Chemistry | 43.00 | **49.00** | 45.00 | **51.00** | Logical Fallacies | 63.80 | **72.39** | 79.14 | **82.21** |
| Coll. Mathematics | 32.00 | **49.00** | 40.00 | **49.00** | Machine Learning | 40.18 | **46.43** | 53.57 | **61.61** |
| Coll. Medicine | 60.69 | **63.58** | 72.83 | **73.99** | Management | 66.99 | **77.67** | **87.38** | 80.58 |
| Coll. Physics | 40.20 | **42.16** | 52.94 | **58.82** | Marketing | 79.49 | **88.03** | 83.76 | **85.04** |
| Computer Security | 68.00 | **77.00** | 72.00 | **76.00** | Medical Genetics | 54.00 | **74.00** | 82.00 | **83.00** |
| Conc. Physics | 54.47 | **64.26** | 77.02 | **80.00** | Miscellaneous | 62.84 | **74.97** | 76.37 | **76.63** |
| Econometrics | 35.96 | **42.11** | 55.26 | **62.28** | Moral Disputes | 51.73 | **67.05** | 63.01 | **71.68** |
| Elec. Eng. | 59.31 | **64.83** | 66.90 | **76.55** | Moral Scenarios | 29.27 | **31.28** | 36.98 | **42.01** |
| Elem. Mathematics | 46.56 | **54.76** | 74.60 | **78.31** | Nutrition | 54.25 | **70.59** | 69.61 | **74.84** |
| Formal Logic | 39.68 | **43.65** | 56.35 | **63.49** | Philosophy | 55.95 | **67.52** | 66.88 | **72.67** |
| Global Facts | 25.00 | **30.00** | **46.00** | 43.00 | Prehistory | 54.32 | **64.51** | 70.68 | **74.38** |
| H.S. Biology | 64.19 | **80.00** | 86.13 | 86.13 | Professional Accounting | 36.88 | **45.39** | 50.00 | **56.74** |
| H.S. CS | 59.00 | **71.00** | 79.00 | **84.00** | Professional Law | 34.68 | **40.94** | 42.44 | **49.02** |
| H.S. Chemistry | **53.69** | 51.72 | **74.88** | 68.47 | Professional Medicine | 46.69 | **61.03** | 68.38 | **77.21** |
| H.S. Eur. Hist. | 67.27 | **74.55** | 78.18 | **80.00** | Professional Psychology | 53.27 | **60.62** | 66.99 | **73.20** |
| H.S. Geography | 62.12 | **78.28** | 79.80 | **82.32** | Public Relations | 52.73 | **58.18** | 62.73 | 62.73 |
| H.S. Gov. & Pol. | 69.95 | **75.65** | 84.97 | **87.05** | Security Studies | 63.67 | **66.53** | 72.24 | **73.88** |
| H.S. Macroeconomics | 58.97 | **64.62** | 73.85 | **76.41** | Sociology | 68.66 | **76.62** | **82.09** | 80.60 |
| H.S. Mathematics | 29.26 | **37.41** | 48.89 | **58.15** | US Foreign Policy | 65.00 | **79.00** | 71.00 | **87.00** |
| H.S. Microeconomics | 63.03 | **75.21** | 82.77 | **87.39** | Virology | 44.58 | **50.00** | 44.58 | **45.18** |
| H.S. Physics | 42.38 | **47.68** | 54.97 | **62.91** | World Religions | 63.74 | **77.19** | 72.51 | **74.85** |
| H.S. Psychology | 73.94 | **82.20** | 86.79 | **87.34** | | | | | |
| **Average** | 53.50 | **62.60** | 67.46 | **71.76** | | | | | |

across all model scales. Specifically, we observe an improvement of **+15.4%** on Qwen3-1.7B (33.78% → 49.18%) and **+14.36%** on Qwen3-4B (51.49% → 65.85%). These gains are particularly significant in algorithmic and symbolic tasks like *Boolean Expressions* and *Web of Lies*, indicating that our method substantially mitigates the reasoning failures often observed in standard SFT models.

## D.7. Training Dynamics

We compare the training stability of our method against the SFT baseline in Figure 6. These experiments were conducted using the Qwen2.5-Math-1.5B model fine-tuned on the NuminaMath dataset. Mathematical reasoning tasks often induce high variance in training gradients, leading to instability. As shown in the figure, the SFT baseline exhibits significant fluctuations and spikes throughout the training process. In contrast, our method maintains a much smoother loss trajectory

*Table 8.* Detailed results on the Big Bench Hard (BBH) benchmark. The table reports the exact match scores for each of the 27 challenging tasks requiring multi-step reasoning. We compare the SFT baseline against our method across Qwen3-1.7B and Qwen3-4B architectures.

| Task | Qwen3-1.7B | | Qwen3-4B | | Task | Qwen3-1.7B | | Qwen3-4B | |
|---|---|---|---|---|---|---|---|---|---|
| | SFT | Ours | SFT | Ours | | SFT | Ours | SFT | Ours |
| Boolean Expressions | 47.60 | **84.00** | 53.20 | **93.20** | Object Counting | 32.80 | **54.40** | **71.20** | 68.00 |
| Causal Judgment | 24.60 | **33.69** | **33.16** | 20.32 | Penguins in a Table | 49.32 | **58.90** | 73.97 | **80.14** |
| Date Understanding | 44.00 | **62.40** | 56.80 | **81.20** | Reasoning Col. Obj. | 51.20 | **69.60** | 68.40 | **88.00** |
| Disambiguation QA | 30.80 | **39.20** | 48.80 | **53.20** | Ruin Names | 32.40 | **44.80** | 40.80 | **62.80** |
| Dyck Languages | 1.60 | **2.40** | 4.80 | **11.20** | Salient Trans. Error Det. | 28.40 | **33.20** | 48.00 | **52.00** |
| Formal Fallacies | 16.00 | **31.60** | 29.60 | **41.60** | Snarks | **50.56** | **50.56** | 55.62 | **60.67** |
| Geometric Shapes | 24.80 | **38.00** | 44.80 | **52.40** | Sports Understanding | **25.60** | 14.00 | 27.20 | **34.80** |
| Hyperbaton | 55.60 | **77.60** | 76.00 | **93.20** | Temporal Sequences | **21.20** | 16.40 | 52.00 | **63.60** |
| Logical Deduction (3) | 54.80 | **76.40** | 86.40 | **92.40** | Tracking Shuffled Obj. (3) | 52.40 | **58.00** | 72.40 | **94.80** |
| Logical Deduction (5) | 35.20 | **40.80** | 42.80 | **64.00** | Tracking Shuffled Obj. (5) | 23.60 | **39.60** | 60.80 | **86.80** |
| Logical Deduction (7) | 20.00 | **33.20** | 32.00 | **50.40** | Tracking Shuffled Obj. (7) | 20.40 | **34.80** | 48.40 | **76.80** |
| Movie Recommendation | 50.00 | **64.80** | 52.80 | **68.80** | Web of Lies | 45.20 | **95.20** | 72.00 | **77.20** |
| Multistep Arithmetic | 23.60 | **72.80** | 47.60 | **82.40** | Word Sorting | 8.40 | **18.00** | 20.40 | **41.20** |
| Navigate | 42.00 | **83.60** | 70.40 | **86.80** | | | | | |
| **Average** | 33.78 | **49.18** | 51.49 | **65.85** | | | | | |

with reduced variance, demonstrating the robustness of our approach on complex mathematical data.

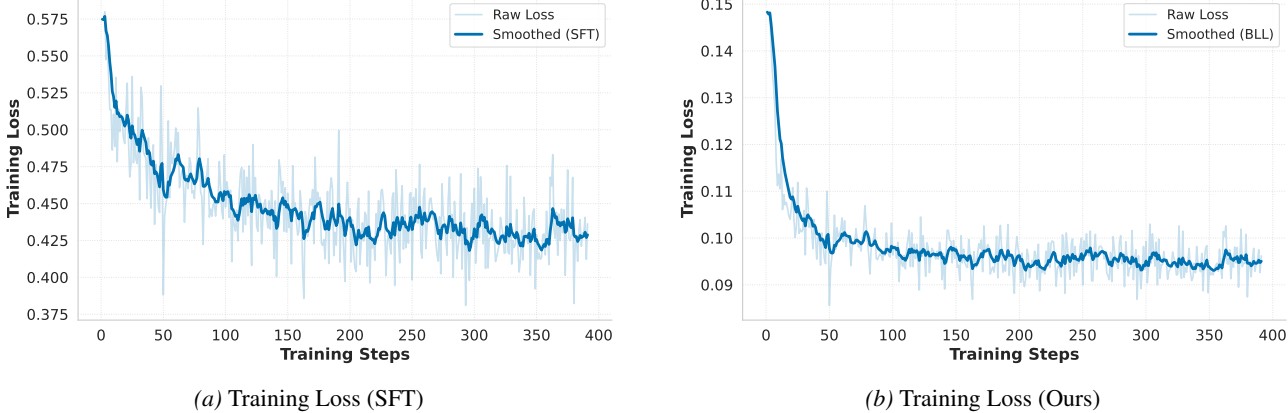

*(a)* Training Loss (SFT)  *(b)* Training Loss (Ours)

*Figure 6.* Comparison of training objectives. (a) The standard **Cross-Entropy loss** trajectory for the SFT baseline. (b) The convergence curve of our proposed **BLL-Loss**. The light blue lines represent raw step-wise loss, while dark blue lines indicate the EMA smoothed trend. Note that the loss scales differ due to the distinct formulations of the objective functions.

# E. Case Studies

To provide a comprehensive understanding of the reasoning behaviors induced by our proposed BLL-loss, we conduct a qualitative analysis across two difficulty levels: GSM8K (standard reasoning) and AIME 2024 (challenging competition-level math).

---

**Question**

Janet's ducks lay 16 eggs per day. She eats three for breakfast every morning and bakes muffins for her friends every day with four. She sells the remainder at the farmers' market daily for $2 per fresh duck egg. How much in dollars does she make every day at the farmers' market?

---

**(Qwen2.5-Math-1.5B-BLL) Response**

Janet's ducks lay 16 eggs per day. She eats 3 eggs for breakfast every morning, so she has $16 - 3 = 13$ eggs left for baking muffins and selling them at the farmers' market.

She bakes muffins for her friends every day with 4 eggs, so she has $13 - 4 = 9$ eggs left to sell.

She sells the remaining eggs at $2 per fresh duck egg, so she makes $9 \times \$2 = \boxed{\$18}$ every day at the farmers' market.

---

**(Qwen2.5-Math-7B-BLL) Response**

Janet's ducks lay 16 eggs per day. She eats 3 eggs for breakfast and uses 4 eggs to bake muffins, so she uses a total of $3 + 4 = 7$ eggs per day.

The number of eggs she has left to sell at the farmers' market is $16 - 7 = 9$ eggs per day.

She sells each egg for $2, so the total amount she makes per day is $9$ eggs $\times \$2/\text{egg} = \$18$.

Therefore, Janet makes $\boxed{\$18}$ every day at the farmers' market.

---

*Figure 7.* Qualitative comparison on a GSM8K math problem.

## E.1. Consistency in Standard Reasoning (GSM8K)

Figure 7 compares the responses of the 1.5B and 7B Qwen2.5-Math models on a standard GSM8K problem. Both models successfully identify variables and formulate correct equations, demonstrating the consistency of BLL-Loss across model scales. While Qwen2.5-Math-1.5B Model (Top) provides a clear logical flow, Qwen2.5-Math-7B Model (Bottom) exhibits enhanced rigor through detailed, step-by-step derivations (e.g., explicit unit conversions). This confirms that BLL-Loss effectively optimizes the reasoning path, generating coherent Chain-of-Thought sequences for standard mathematical tasks.

## E.2. Generalization on Challenging Tasks (AIME 2024)

Figure 8 demonstrates the model's generalization capability on challenging AIME 2024 problem. Despite the significant complexity gap compared to the training data (NuminaMath), the model trained with BLL-loss maintains strict logical coherence. It successfully identifies core constraints and systematically decomposes the problem, avoiding common hallucinations. This confirms that our method effectively enhances robustness in handling unseen, high-difficulty mathematical domains.

---

**Question**

Every morning Aya goes for a 9-kilometer-long walk and stops at a coffee shop afterwards. When she walks at a constant speed of $s$ kilometers per hour, the walk takes her 4 hours, including $t$ minutes spent in the coffee shop. When she walks $s + 2$ kilometers per hour, the walk takes her 2 hours and 24 minutes, including $t$ minutes spent in the coffee shop. Suppose Aya walks at $s + \frac{1}{2}$ kilometers per hour. Find the number of minutes the walk takes her, including the $t$ minutes spent in the coffee shop.

---

**(Qwen2.5-Math-7B-BLL) Response**

1. **Define the variables:** Let $s$ be Aya's walking speed (km/h) and $t$ be the time (min) spent in the coffee shop. Total times: 4 hours at speed $s$; 2 hours 24 minutes at speed $s + 2$.

2. **Convert times to hours:**
   - 4 hours is already in hours.
   - 2 hours 24 minutes $= 2 + \frac{24}{60} = 2.4$ hours.

3. **Set up the equations:**
   - At speed $s$: $\dfrac{9}{s} + \dfrac{t}{60} = 4$                                  (1)
   - At speed $s + 2$: $\dfrac{9}{s + 2} + \dfrac{t}{60} = 2.4$                     (2)

4. **Solve the system of equations:**

   - Subtract (2) from (1):
     $$\frac{9}{s} - \frac{9}{s + 2} = 1.6$$
   - Simplify:
     $$\frac{18}{s(s + 2)} = 1.6$$
   - Rearrange to quadratic:
     $$1.6s^2 + 3.2s - 18 = 0$$

   - Quadratic formula:
     $$s = \frac{-3.2 \pm \sqrt{3.2^2 + 4 \cdot 1.6 \cdot 18}}{2 \cdot 1.6}$$
     $$s = \frac{-3.2 \pm 11.2}{3.2}$$
   - Positive root:
     $$s = 2.5 \text{ km/h}$$

5. **Find $t$:** Substitute $s = 2.5$ into eq (1):

   $$\frac{9}{2.5} + \frac{t}{60} = 4 \implies 3.6 + \frac{t}{60} = 4 \implies \frac{t}{60} = 0.4 \implies t = 24 \text{ min}$$

6. **Calculate final time:**
   - New speed: $s + 0.5 = 2.5 + 0.5 = 3$ km/h.
   - Walking time: $\dfrac{9}{3} = 3$ hours.
   - Total time: 3 hours + 24 minutes = 3.4 hours.
   - In minutes: $3.4 \times 60 = 204$ minutes.

**Conclusion:**

$$\boxed{204}$$

---

*Figure 8.* Case study on a AIME2024 math problem.

