# OpenReview forum: "Don't Force the Fit: Bounded Log-Likelihood Loss for Enhanced Reasoning in Large Language Models"
_ICML.cc/2026/Conference — ICML 2026 spotlight_

### Official Review · Reviewer_VWEE · 2026-03-04

**Soundness:** 3
**Presentation:** 3
**Significance:** 3
**Originality:** 3
**Overall Recommendation:** 4
**Confidence:** 4

**Summary:**

This work points out that the current SFT forces LLMs to fit all response tokens of the training set.
However, in reasoning tasks where correctness is defined by logical validity or final outcomes rather than exact token realizations, this assumption can lead to optimization misalignment.
To address this problem, the authors propose a simple solution, BLL-Loss, that bounds gradient contributions from low-probability tokens while preserving conventional optimization behavior. Experiments show that BLL-Loss improves reasoning generalization across diverse model scales and challenging benchmarks.

**Compliance With Llm Reviewing Policy:**

Affirmed.

**Final Justification:**

I think this is a good job, but its performance improvement over DFT is too small. Therefore, I have decided to keep the original positive score of 4.

**Key Questions For Authors:**

see weakness

**Limitations:**

yes

**Strengths And Weaknesses:**

**Strengths**
- Unlike hard-masking strategies that require careful tuning of discrete probability thresholds, BLL-Loss is a smooth, fully differentiable, and parameter-free objective that naturally neutralizes outlier gradients.
- The method consistently outperforms standard SFT and baselines like NEFT and GEM across mathematical problem solving, code generation, and general reasoning tasks.
- Clear theoretical derivation and proof
- This can provide a new insight into SFT training.

**Weaknesses**
- The premise that "not all tokens are equally important" is not entirely novel, and existing literature has explored this extensively. For instance, DFT[1] shares a very similar intuition regarding down-weighting low-probability tokens. Furthermore, CFT[2] demonstrates that SFT training is not necessary for all tokens. The paper fails to discuss CFT and DFT in its related work and omits DFT as a direct baseline in the main experiments.

- The method's effectiveness on inherently weak base models is not rigorously proven. While the authors attempt to address scalability to weaker models in Table 5 using Llama-3.2-1B, the evaluation is flawed. Table 5 only compares BLL-Loss against the Base model and standard SFT. To prove its robustness, it should have included the optimization baselines from the main experiments (NEFT, GEM), as well as competitive methods like DFT.


[1]On the Generalization of SFT: A Reinforcement Learning Perspective with Reward Rectification

[2]Enhancing Large Language Model Reasoning via Selective Critical Token Fine-Tuning

---

> ### Author Rebuttal · Authors · 2026-03-31
>
> We thank the reviewer for directing us to DFT and CFT. We agree that contextualizing our method alongside these highly relevant works strengthens our paper, and we will include a comprehensive discussion of them in the revised manuscript. Below, we address your concerns regarding novelty and empirical baselines.
>
> # Response to W1: How BLL differs from DFT and CFT
> - While DFT approaches this issue from an RL and reward rectification perspective, it lacks a granular, token-level investigation. It does not explore the semantic roles of low-probability tokens in mathematical reasoning, nor does it analyze the direct impact of masking them. Our work bridges this gap by providing a detailed empirical diagnosis, demonstrating that these low-probability tokens are predominantly task-irrelevant stylistic artifacts.
>
> - CFT relies on counterfactual perturbations to identify "critical tokens." This heuristic approach introduces immense computational overhead, as importance scores must be recalculated for every new model and dataset. In contrast, we demonstrate that a model's inherent pre-trained prior is sufficient to dynamically assess token importance. BLL is a completely parameter-free objective that scales seamlessly across diverse architectures without requiring external perturbations or manual tuning.
>
> Due to the limited time in the rebuttal period, we evaluated BLL against both the DFT baseline and the p-objective (suggested by Reviewer HkPE) using Qwen3-1.7B and Qwen3-4B on a randomly sampled 10k subset. We would like to highlight that the core contribution of our work is a complete, theoretically grounded pipeline: an empirical diagnosis of low-probability artifacts, followed by the derivation of a novel, parameter-free loss function from two entirely distinct mathematical perspectives. Beyond these valuable theoretical insights, the empirical results below also demonstrate that our method performs better than the baselines.
>
> | Methods         | GSM8K | Math500 | Gaokao | Minerva | Olympiad |  AMC  | AIME | Avg.  |
> |:----------------|:-----:|:-------:|:------:|:-------:|:--------:|:-----:|:----:|:-----:|
> | P(Qwen3-1.7B)   | 76.76 |  54.88  | 47.12  |  17.83  |  20.32   | 29.06 | 0.83 | 35.26 |
> | DFT(Qwen3-1.7B) | 76.43 |  55.90  | 47.74  |  16.71  |  21.19   | 29.84 | 1.66 | 35.64 |
> | BLL(Qwen3-1.7B) | 75.15 |  55.53  | 47.56  |  17.94  |  21.86   | 26.88 | 4.99 | 35.70 |
> | P(Qwen3-4B)     | 88.63 |  64.16  | 54.77  |  22.57  |  29.04   | 31.41 | 8.33 | 42.70 |
> | DFT(Qwen3-4B)   | 85.71 |  63.11  | 55.34  |  23.59  |  30.26   | 31.56 | 6.25 | 42.26 |
> | BLL(Qwen3-4B)   | 88.64 |  64.09  | 55.72  |  18.05  |  30.19   | 35.31 | 9.17 | 43.02 |
>
> # Response to W2: Rigorous Baselines on Weak Models
>
> We thank the reviewer for pointing out this gap. We agree that evaluating smaller-scale models requires strong baselines. Accordingly, we expanded our evaluation on Llama-3.2-1B to include NEFT, GEM, and the newly suggested DFT.
> As shown below, BLL-Loss achieves the highest average score (4.38), nearly doubling the performance of standard SFT (2.41) and performing better than the highly competitive DFT baseline. Notably, on tasks like AMC, BLL demonstrates a clear advantage over DFT (2.66 vs. 1.56). We hope these comprehensive results address your concern, demonstrating that the effectiveness and robustness of BLL-Loss hold true against competitive baselines, even on weak base models.
>
> | Methods        | GSM8K | Math500 | Gaokao | Minerva | Olympiad | AMC  | AIME |   Avg.   |
> |:---------------|:-----:|:-------:|:------:|:-------:|:--------:|:----:|:----:|:--------:|
> | BASE           | 1.22  |  1.78   |  3.38  |  0.84   |   0.98   | 1.41 |  0   |   1.37   |
> | SFT            | 4.64  |  2.95   |  5.41  |  0.79   |   1.20   | 1.88 |  0   |   2.41   |
> | GEM            | 0.91  |  1.05   |  3.17  |  0.61   |   0.64   | 1.09 |  0   |   1.07   |
> | NEFT           | 4.83  |  3.10   |  5.87  |  1.07   |   1.40   | 2.03 |  0   |   2.61   |
> | DFT            | 10.18 |  5.48   |  8.86  |  2.17   |   1.16   | 1.56 |  0   |   4.20   |
> | **BLL (Ours)** | 10.41 |  5.51   |  8.27  |  2.01   |   1.79   | 2.66 |  0   | **4.38** |

---

> > ### Author Rebuttal · Reviewer_VWEE · 2026-04-01
> >
> > I think this is a good job, but its performance improvement over DFT is too small. Therefore, I have decided to keep the original positive score of 4.

---

### Official Review · Reviewer_QH87 · 2026-03-12

**Soundness:** 4
**Presentation:** 4
**Significance:** 3
**Originality:** 4
**Overall Recommendation:** 5
**Confidence:** 4

**Summary:**

The paper presents a novel bounded log-likelihood loss (BLL-Loss) for supervised fine-tuning for reasoning in LLMs. The method is motivated by an observation that low-probability tokens in reasoning demonstrations often correspond to task-irrelevant stylistic artifacts, and reducing their gradients in SFT could consistently improve the performance. Based on this observation, the authors argue that SFT for reasoning LLMs should not treat all tokens equally during supervision. The proposed loss could be derived from the conventional cross-entropy loss by both the alternative series and the log-odd decomposition, and the gradient analysis reveals that the BLL-Loss adaptively suppresses the gradient for low-probability tokens. Experiments on mathematical problems, code generation, and general reasoning demonstrate clear and significant improvement in accuracy.

**Compliance With Llm Reviewing Policy:**

Affirmed.

**Key Questions For Authors:**

1. The pre-experiments observe that low-probability tokens usually correspond to task-irrelevant stylistic artifacts. (1) Does this pattern generalize to other tasks? (2) Since stylistic phrases often appear in specific parts of responses (e.g., opening or closing), how does the likelihood of tokens vary with token positions? If so, could it be leveraged to further improve the proposed method?
2. Does the boundedness of the proposed loss weaken the learning signal for hard tokens?
3. What is the threshold for gradient clipping in the ablation studies? Is the proposed method still superior to gradient clipping with the best threshold?

**Limitations:**

Yes

**Strengths And Weaknesses:**

**Strengths:**

1. The paper is well-organized, fluent in logic, and provides detailed and rigorous mathematical explanations.
2. The authors presented two converted divergences, the alternating series and the log-odd decomposition, to support the rigidity in the BLL-loss design.
3. A further theoretical analysis of the gradient of the BLL loss reveals that the backward propagation of BLL-loss actually modulates the original CE loss to suppress the gradient for low-probability tokens.
4. The key idea is concise but powerful. The proposed loss is parameter-free and explainable, indicating strong potential for adaptation in real industrial training scenarios.
5. Sufficient main experiments and ablation studies are provided to support the claim.

**Weaknesses**

1. The pre-experiment in token likelihood analysis (Section 2.3) is only conducted on general math problem solving. It would be more interesting to see the word cloud for other tasks, such as code generation and general reasoning.
2. Label smoothing is discussed in the related works, but not included as a baseline in the main experiments. Similarly, the token masking technique discussed in the pre-experiments in Section 2.4 is also not compared in the main experiments.
3. Only the gradients of the target tokens are discussed in Section 3.3. However, the gradients for non-target tokens are also informative, which could reveal how token likelihoods for alternative tokens evolve during fine-tuning.

---

> ### Author Rebuttal · Authors · 2026-03-31
>
> We thank the reviewer for the thorough and constructive feedback. To address your comments comprehensively, we have grouped related questions by theme:
> # Response to (W1, Q1.1): Generalization across Tasks
> Regarding generalization to other domains, our extended evaluations (Section 4.5, Table 3) show that BLL-Loss significantly outperforms standard SFT on code generation and general reasoning tasks. This empirical success suggests that the "forced fit" problem, which is prevalent in mathematical reasoning, likely extends to these domains as well. To further analyze these distributions, we will include token word clouds for both code generation and general reasoning in the appendix of the revised manuscript.
> # Response to Q1.2: Positional Variations
> Stylistic phrases do indeed tend to cluster at the beginning and end of responses. While we could introduce positional heuristics (e.g., positional decay weights) to address this, a core strength of BLL is its dynamic, position-agnostic nature. It automatically bounds the gradient for stylistic tokens based strictly on the model's confidence, effectively eliminating the need for manual, position-based rules.
> # Response to W2: Baseline Comparisons
> We initially omitted hard masking because its optimal threshold fluctuates drastically across datasets, making it inflexible. To address your suggestion, we evaluated Label Smoothing (LS) and token masking (p<0.15 and p<0.5). The results below perfectly validate our motivation:
> - LS is ineffective: Global penalties barely improve upon SFT (32.12 vs. 31.72 Avg), failing to capture the strict deterministic constraints of reasoning tasks.
> - Masking is highly sensitive to thresholds: Performance fluctuates significantly depending on the chosen cutoff, making a universally optimal setting elusive (e.g., $p<0.5$ favors GSM8K, while $p<0.15$ favors Minerva).
> - BLL is robust: BLL-Loss achieves the highest overall average (46.41) completely parameter-free. By dynamically adapting to token-level confidence, it bypasses the need for manual tuning and outperforms fixed thresholds.
>
> | Methods         | GSM8K | Math500 | Minerva | Olympiad |  AMC  |   Avg.    |
> |:----------------|:-----:|:-------:|:-------:|:--------:|:-----:|:---------:|
> | Standard SFT    | 72.61 |  43.26  |  11.86  |  13.82   | 17.03 |   31.72   |
> | Label Smoothing | 70.90 |  42.95  |  13.78  |  14.70   | 18.28 |   32.12   |
> | Masking(p<0.15) | 80.88 |  61.50  |  22.96  |  27.55   | 34.53 |   45.49   |
> | Masking(p<0.5)  | 81.40 |  64.20  |  21.23  |  27.16   | 34.06 |   45.61   |
> | BLL (Ours)      | 81.31 |  64.66  |  22.96  |  26.69   | 36.41 | **46.41** |
>
> # Response to (W3,Q2): Non-target Tokens (W3) & Hard Tokens (Q2)
> - Non-target Tokens: We appreciate this theoretical insight, which highlights a key advantage of our method. For  non-target tokens ($j \\neq t$). Under Cross-Entropy (CE), the gradient with respect to a non-target logit is $\\nabla\_{z\_j} \\mathcal{L}\_{CE} = p\_j$.
> When the model confidently predicts a semantically valid token ($p\_j \\to 1$) but the ground truth is a low-probability stylistic token ($p\_t \\to 0$), CE induces a strong "push-away" effect, forcing the model to unlearn valid semantic representations.
> In contrast, BLL introduces a dynamic modulation: $\\nabla\_{z\_j} \\mathcal{L}\_{BLL} = \\lambda(p\_t) \\nabla{z\_j} \\mathcal{L}\_{CE}$, where $\\lambda(p\_t)=\\frac{p\_t}{2-p\_t}$. This demonstrates that BLL substantially mitigates the push-away effect in high-surprisal stylistic regions, preventing destructive updates on valid tokens and better preserving the model's reasoning capability.
> - Hard Tokens: While BLL mathematically bounds the magnitude of the gradient for low-probability tokens, it does not detrimentally "weaken" the learning of genuine reasoning steps. Instead, it regularizes the learning signal to prevent optimization collapse, leading to better generalization on hard tasks.
> # Response to (Q3): Gradient Clipping Thresholds
> We appreciate the opportunity to clarify and expand upon this. In the main paper (Section 4.3, Table 2), we reported tight gradient clipping thresholds of 0.3 and 0.5.
> To further examine the effect of threshold selection, we additionally tested a threshold of 0.8. The results are presented in the table below,
> and it can be observed that, under all tested clipping thresholds, the performance of the model does not surpass that of our proposed method.
>
> | Methods         | GSM8K | Math500 | Minerva | Olympiad |  AMC  |   Avg.    |
> |:----------------|:-----:|:-------:|:-------:|:--------:|:-----:|:---------:|
> | Standard SFT    | 72.61 |  43.26  |  11.86  |  13.82   | 17.03 |   31.72   |
> | Grad Clip (0.3) | 70.43 |  42.89  |  15.66  |  14.90   | 16.56 |   32.09   |
> | Grad Clip (0.5) | 70.58 |  42.86  |  16.48  |  15.25   | 15.47 |   32.13   |
> | Grad Clip (0.8) | 70.67 |  42.24  |  17.19  |  16.06   | 16.41 |   32.51   |
> | BLL (Ours)      | 81.31 |  64.66  |  22.96  |  26.69   | 36.41 | **46.41** |

---

> > ### Author Rebuttal · Reviewer_QH87 · 2026-04-02
> >
> > Q2: The authors claimed that bounded loss prevents "optimization collapses" and leads to "better generalization on hard tasks" without further illustration through literature, reasoning, or experiments.
> > W1: As is also pointed out by Reviewer HkPE, the evidence in Investigation 1 supporting the basis claim is based on only one task. I wish to see stronger support in other fields in the final version, as is promised by the authors.

---

> > > ### Author Response · Authors · 2026-04-03
> > >
> > > # Mechanism and Empirical Evidence of BLL
> > > Due to the response length limit, we were unable to fully elaborate on this point in our previous reply. We provide a more detailed explanation below.
> > >
> > > - **Preventing optimization collapse**: BLL adaptively scales the gradient based on the model’s confidence for each token. For hard tokens, the corresponding gradients are significantly suppressed. This mechanism prevents the training process from being dominated by updates from low-probability tokens, thereby reducing overfitting to unstable signals. This observation is further supported by our low-probability token masking experiments: removing these signals leads to improved reasoning performance, indicating that suppressing low-probability tokens helps stabilize optimization and improve overall performance.
> > >
> > > - **Better generalization on hard tasks**: This phenomenon is observed in our main experiments (Table 1). We focus on the results from Qwen2.5-Math-1.5B for analysis, while noting that similar trends are consistently observed across other models.
> > > As shown in the table, standard SFT significantly improves performance on simple benchmarks such as GSM8K and Math500. However, on challenging benchmarks, including Olympiad and AMC, SFT lead to performance degradation, whereas our method consistently achieves substantial improvements.
> > > Specifically, BLL achieves notable gains on Olympiad (+10.72), and AMC (+18.60), demonstrating its strong advantage on hard reasoning tasks.
> > >
> > > | Methods | GSM8K | Math500 | Gaokao | Minerva |    Olympiad    |      AMC       | AIME |  Avg.  |
> > > |:--------|:-----:|:-------:|:------:|:-------:|:--------------:|:--------------:|:----:|:------:|
> > > | BASE    | 31.06 |  31.90  | 24.26  |  8.01   |     15.97      |     17.81      | 2.49 | 18.79  |
> > > | SFT     | 72.61 |  43.26  | 39.11  |  11.86  | 13.82 (-2.15)  | 17.03 (-0.78)  | 0.83 | 28.36  |
> > > | BLL     | 81.31 |  64.66  | 53.28  |  22.96  | 26.69 (+10.72) | 36.41 (+18.60) | 5.62 | 41.56  |
> > >
> > > #  Additional Empirical Evidence
> > >
> > > We agree that extending the analysis to additional reasoning tasks is important for a more comprehensive evaluation. In the current version, we initially focus on math tasks due to their clear reasoning structure, which facilitates visualization through word clouds. In the final version, we will extend our experiments to include code and general reasoning tasks, providing stronger and more comprehensive empirical evidence to support our claims.

---

### Official Review · Reviewer_HkPE · 2026-03-13

**Soundness:** 2
**Presentation:** 3
**Significance:** 2
**Originality:** 2
**Overall Recommendation:** 4
**Confidence:** 4

**Summary:**

The authors propose a new loss function--the Bounded Log-Likelihood (BLL) objective--whose gradient contribution is suppressed for low-probability tokens. They motivate and derive this objective from two perspectives: series convergence and log-odds decomposition. Subsequently, they empirically validate the effectiveness of the proposed objective in math and general reasoning domains using Qwen and LLaMA models.

**Compliance With Llm Reviewing Policy:**

Affirmed.

**Final Justification:**

I find the motivation of this work to be quite novel; however, it appears to be a reweighted p-objective, despite the different derivation presented in the paper. Furthermore, the distinction between the p-objective and DFT is marginal. Therefore, I have decided to maintain my original score of 4.

**Key Questions For Authors:**

1. The paper states that Eq. (10) is a generalized version of Eq. (9), but if $\beta \in [0,1]$, it seems that Eq. (9) is not actually recovered as a special case of Eq. (10). I would appreciate it if the authors could clarify this point.

2. Could the authors also include a comparison with the p-objective [1]? Since the p-objective, which is a variant of the GCE loss, also suppresses gradients on low-probability tokens by construction, such a comparison would be highly relevant.

3. In the main results, have the authors considered why the performance gap between the BLL loss and the standard SFT loss appears to widen as model size increases? I think this could lead to an interesting and informative analysis.

#### References
[1] Li, Gaotang, et al. "Beyond log likelihood: Probability-based objectives for supervised fine-tuning across the model capability continuum." arXiv preprint arXiv:2510.00526 (2025).

**Limitations:**

The authors discussed the limitations and the societal impact. It would be better to provide a slightly more detailed discussion of the potential negative societal impacts.

**Strengths And Weaknesses:**

### Strengths

1. The paper provides an accessible explanation of why token-level SFT can potentially distort the optimization trajectory, and in particular clarifies the role played by low-probability tokens (even though this phenomenon is already known).
2. It empirically validates the effectiveness of the BLL loss objective in math and general reasoning domains.
3. The main results demonstrate improved performance compared to standard SFT.

### Weaknesses

1. The conclusion drawn in Investigation 1 may largely depend on the choice of base models and datasets. In the experiments (Section 2.3), the authors use the NuminaMath CoT dataset and Qwen2.5-Math-1.5B as the base model. Since this model is knowledgeable about the NuminaMath CoT dataset, observing realization-level mismatch for low-probability tokens is, in some sense, expected. The conclusion could differ when the base model is evaluated on datasets for which it is not knowledgeable. Therefore, drawing a general conclusion as in Section 2.3 based on this specific case may constitute a premature generalization.
2. Additional experiments are needed to sufficiently verify whether this objective remains robust in domains where the model prior is weak. For example, further validation on model-weak end tasks--such as the datasets reported in Table 4 of [1]--would be necessary to support the claimed robustness of this objective.

#### References
[1] Li, Gaotang, et al. "Beyond log likelihood: Probability-based objectives for supervised fine-tuning across the model capability continuum." arXiv preprint arXiv:2510.00526 (2025).

---

> ### Author Rebuttal · Authors · 2026-03-31
>
> We sincerely thank the reviewer for the insightful feedback. We address your detailed comments point-by-point below:
> #  Response to W1: Premise of Investigation 1
>  We fully agree that the empirical observations in Investigation 1 rely on the base model possessing a reasonable prior for the target domain. In fact, as discussed in Section 2.2, our work builds upon the core hypothesis of LIMA, which posits that knowledge is primarily acquired during pre-training, and post-training serves merely to elicit this knowledge. This foundational premise inherently implies that the post-training dataset must reasonably align with the model's pre-trained distribution, rather than acting as a vehicle for injecting completely novel knowledge. If the dataset distribution is entirely disjoint from the model's prior, even core reasoning tokens would exhibit high surprisal. We will explicitly clarify this boundary condition in the revised manuscript.
> # Response to W2: Experiments on [1]'s Datasets
> To address your suggestion, we conducted additional experiments during the rebuttal period using the specific datasets reported in Table 4 of [1]. We evaluated  our BLL-Loss on the Llama-3.1-8B model. The new empirical results are presented below:
>
> | Methods   | Train Size | Test Size | Score |
> |:----------|:-----------|:----------|:------|
> | BLL       | 40K        | 40K       | 0     |
>
> As anticipated in our prior discussion (Response to W1), BLL underperforms in this specific setting, consistent with the observations reported in [1]. Because this dataset represents a severe out-of-distribution (OOD) shift from the model's pre-training prior, down-weighting high-surprisal tokens inherently hinders the model's ability to memorize and acquire entirely novel core concepts.
> These results explicitly define the boundary conditions of our method: BLL is highly effective for in-distribution reasoning elicitation, but is not intended for severe OOD knowledge injection. Developing a unified objective function that seamlessly handles both scenarios remains a promising direction for future work. We will include these additional results and a discussion of this limitation in the revised manuscript.
>
> # Response to Q2: New  Baselines
> We appreciate the reference to this relevant concurrent work. We evaluated the p-objective and DFT (Reviewr VWEE) on a 10k subset using Qwen3-1.7B and 4B. As shown below, BLL-Loss achieves the highest average performance and demonstrates two distinct advantages:
>
> - BLL consistently excels on the most challenging benchmarks across different model scales. For instance, on the highly difficult AIME dataset, BLL achieves 4.99 with the 1.7B model (vs. P's 0.83) and reaches 9.17 with the 4B model (vs. P's 8.33). This confirms that our rigorous loss formulation provides tangible benefits for complex reasoning tasks.
> - Unlike the p-objective, which we observed to fluctuate around negative values, BLL-Loss remains strictly positive with a smooth, monotonically decreasing trajectory (similar to standard CE). This ensures a far more stable and easily diagnosable training process.
>
> | Methods         | GSM8K | Math500 | Gaokao | Minerva | Olympiad |  AMC  |   AIME   |   Avg.    |
> |:----------------|:-----:|:-------:|:------:|:-------:|:--------:|:-----:|:--------:|:---------:|
> | P(Qwen3-1.7B)   | 76.76 |  54.88  | 47.12  |  17.83  |  20.32   | 29.06 |   0.83   |   35.26   |
> | DFT(Qwen3-1.7B) | 76.43 |  55.90  | 47.74  |  16.71  |  21.19   | 29.84 |   1.66   |   35.64   |
> | BLL(Qwen3-1.7B) | 75.15 |  55.53  | 47.56  |  17.94  |  21.86   | 26.88 | **4.99** | **35.70** |
> | P(Qwen3-4B)     | 88.63 |  64.16  | 54.77  |  22.57  |  29.04   | 31.41 |   8.33   |   42.70   |
> | DFT(Qwen3-4B)   | 85.71 |  63.11  | 55.34  |  23.59  |  30.26   | 31.56 |   6.25   |   42.26   |
> | BLL(Qwen3-4B)   | 88.64 |  64.09  | 55.72  |  18.05  |  30.19   | 35.31 | **9.17** | **43.02** |
>
> We will include these empirical results and discussions in the final version.
> # Response to Q3:The Scaling  of the Performance
> We thank the reviewer for raising this question. However, we observe that the absolute performance gains remain consistent across varying model sizes. For instance, across the Qwen3 series (1.7B, 4B, 8B), the gap stabilizes at +9.44, +11.78, and +10.34.  We conjecture that the proportion of stylistic and non-reasoning outlier tokens remains constant, leading to only marginal differences in the final performance.

---

> > ### Author Rebuttal · Reviewer_HkPE · 2026-04-03
> >
> > Thank you to the authors for the rebuttal. I have the following comments:
> >
> > 1. Could you please provide a more detailed experimental setup regarding the response to Q2 on the new baselines?
> >
> > 2. While reading the paper, I found that this limitation was not clearly articulated. As demonstrated by the experiment presented in Response W2, the proposed loss appears to have a distinct limitation. Therefore, I would recommend revising the manuscript to make this limitation more explicit, rather than simply stating that the approach "builds upon the hypothesis of LIMA."

---

> > > ### Author Response · Authors · 2026-04-03
> > >
> > > # Experimental Setup for New Baseline
> > >
> > > - Due to time and computational constraints, we construct a reduced training set by randomly sampling 10K instances from the datasets used in our original paper. For training, we adopt Qwen3-1.7B-Base and Qwen3-4B-Base as the backbone models.
> > >
> > > - All training configurations follow the settings described in the main paper. We implement all experiments using the LLaMA-Factory framework, training for 1 epoch with batch size 256, learning rate
> > > 1.0e-5, and bfloat16 precision.
> > >
> > > - For evaluation, we adopt the open-source evaluation pipeline from Qwen-2.5-Math, where correctness is determined by matching the numerical answer within the \boxed{} format. To ensure robustness and reduce variance, each result is averaged over 16 independent runs.
> > >
> > > #  Clarification and Discussion of Limitations
> > > We agree that this limitation is not clearly stated in the current version and will revise the final version to make it explicit. Specifically:
> > >
> > > - We further clarify that our study focuses on how post-training can better elicit and leverage capabilities acquired during pre-training, rather than introducing entirely new knowledge.
> > > - We explicitly state  that our method relies on a reasonable overlap between the pre-training distribution and the post-training data. Under this condition, BLL can more effectively elicit and leverage capabilities acquired during pre-training. However, when the overlap is negligible and the task requires incorporating new knowledge, BLL  become ineffective, in which case standard SFT is more suitable.
> > > - We will also include experimental results on the dataset in [1] to explicitly show the performance of BLL on this dataset, providing a more direct illustration of its limitations.

---

### Official Review · Reviewer_gM9i · 2026-03-21

**Soundness:** 3
**Presentation:** 4
**Significance:** 3
**Originality:** 3
**Overall Recommendation:** 5
**Confidence:** 4

**Summary:**

Reasoning tokens from expert demonstration are just one semantic realization choice while other tokens can express the same reasoning content. In this situation, the paper argues that SFT on reasoning traces penalizes the model for different semantic choices which arise from prior structures learned during pre-training that are equally valid. Further, the paper argues that such semantic differences usually make up the low-probability regions leading to outsized gradient effects. Thus. they go on to provide a bounded log loss likelihood method that limits the influence of these low-probability tokens while preserving the MLE behavior on high confidence regions. They show impressive results on reasoning benchmarks.

**Compliance With Llm Reviewing Policy:**

Affirmed.

**Key Questions For Authors:**

1. I brought this up in the assessment as well, repeating here as a question. While the BLL method reduces impact of low-probability tokens, does it actually only reduce penalization of semantic tokens? I understand the demonstration from Investigation-1 but the scale and thoroughness of that investigation does not extend to the rigor of the actual proposed framework and I would like to understand that part more.

2. Could the authors zoom into the differences in landscape between their method and gradient clipping? They mention a different landscape and show improvements over gradient clipping, yet intuitively they feel similar since large gradient norms do arise from low probability tokens and get clipped to some constant.

3. While it is entirely up to the authors to pick a setting of interest, such as reasoning traces, I wonder if this problem is also persistent in standard language modeling tasks? For instance abstractive summarization? Is it possible there are related works in those fields that could address this question already?

**Limitations:**

yes

**Strengths And Weaknesses:**

The paper is well-written, promptly addresses limitations and identifies possible oversights and gaps in experiment construction. For example, the naive thresholding on the token probability provided was immediately recognized to be adhoc and not suited for a principled framework that practitioners can generally apply. I also enjoyed the key takeaways portion of the presentation.

The paper is technically sound, providing simple theoretical design to provide intuition for their proposed methodology. While not exactly a weakness, my perspective on the provided theory is that it is more assistive than governing. That is to say, it provides intuition pumps that helps a reader wrap their head around the proposed loss objective but is not necessarily a governing force that solves for an optimization of semantic-only tokens. Another way to phrase this: there is an intuition that low-probability tokens are concerned with semantic / stylistic differences and then there is the theory about bounding the gradients from such low-probability tokens, but closing the gap would require also incorporating that intuition into the theory, which does feel admittedly quite hard.

The bounded log likelihood approach is novel (to the best of my knowledge) and the paper serves to provide deeper insights into untangling priorities in SFT. The experimental results are strong and given the current widespread adoption of language models, its impact could be high.

Overall, this is a well written paper providing valuable gains in performance from a simple methodology modification.

---

> ### Author Rebuttal · Authors · 2026-03-31
>
> We sincerely thank the reviewer for the constructive feedback, the recognition of our theoretical contributions, and the positive evaluation. We address each question below.
> # Response to Q1: Mechanism of BLL on Semantic vs. Stylistic Tokens
> - We appreciate this insightful question, which touches on the core mechanism of our framework. From a purely mathematical standpoint, BLL does not exclusively target stylistic tokens. Because it operates entirely based on the model's confidence (predicted probability), BLL systematically bounds the gradient for any low-probability token, regardless of its semantic role.
>
> - A natural concern is that low-probability tokens may correspond to critical reasoning steps, and downweighting them could harm overall performance.
> However, our empirical results suggest otherwise: BLL acts as a robust regularization mechanism rather than a limitation. As shown in our main results (Table 1), on highly challenging competition math tasks involving complex reasoning, standard SFT degrades performance, whereas BLL achieves substantial improvements.
> We attribute this to the behavior of standard SFT when the target probability approaches zero, which induces excessively large gradient updates. This encourages the model to overfit intractable hard examples, wasting representational capacity and potentially distorting the underlying reasoning structure.
>
> # Response to Q2: Optimization Landscape
> We thank the reviewer for highlighting this distinction. The key difference lies in taking smaller steps toward noisy targets (gradient clipping) versus dynamically disengaging from noisy targets altogether (BLL).
> - Under the standard Cross-Entropy landscape, as the target probability $\pi_\theta(y_t|x) \to 0$, the gradient with respect to the logit converges to a maximal constant magnitude of $1$. Gradient clipping merely scales down the global step size. The underlying loss landscape retains its singularity, and the objective continues to exert a persistent, maximal demand on the model to fit this noisy or stylistic token. Consequently, the model is still forced to allocate representational capacity to it, albeit at a slower learning rate.
> - In contrast, BLL fundamentally reshapes the loss surface. As derived in Eq. (13) and Eq. (14), BLL introduces a dynamic modulation factor $\lambda(\pi_\theta) = \pi_\theta / (2 - \pi_\theta)$. Crucially, as $\pi_\theta \to 0$, the BLL gradient vanishes to zero ($\lim_{\pi_\theta \to 0} \|\nabla \mathcal{L}_{BLL}\| = 0$). This creates a "soft-screening" mechanism: instead of stubbornly fitting intractable hard samples, the gradient naturally detaches, allowing the model to completely ignore realization-specific artifacts.
> - Furthermore, gradient clipping is a post-hoc, global heuristic applied to the entire batch's gradient norm, penalizing all tokens blindly if the cumulative norm exceeds a threshold. BLL, however, is a localized, token-level geometric regularization that intrinsically modulates the supervisory signal based on the model's explicit confidence for each specific token.
> # Response to Q3:  Applicability to Standard Language Modeling
> We appreciate the reviewer raising this thought-provoking point. The issue of strict MLE forcing models to fit arbitrary surface realizations is a well-known challenge in standard language modeling and open-ended generation (often referred to as the "one-to-many" mapping problem).
> - Several related techniques have been developed to relax the rigid, token-level matching of standard SFT. Notable examples include Label Smoothing  (to prevent over-confidence) and recent diversity-promoting objectives like GEM. We actually included GEM as a baseline in our main experiments (Table 1), demonstrating our consideration of general SFT regularization methods.
> - In tasks like summarization, relaxing the MLE objective aims to increase linguistic diversity. In mathematical reasoning, however, correctness is strictly defined by logical validity. General regularization methods like Label Smoothing apply a uniform, global penalty, which can inadvertently blur the strict deterministic constraints required for step-by-step logical deduction.
> - While token-level overfitting exists across domains, addressing it in reasoning tasks requires a structurally distinct approach. Nevertheless, the core mechanism of BLL holds significant potential to be extended to general LLMs. Evaluating whether our confidence-based filtering benefits broader open-ended generation, such as abstractive summarization, is an exciting avenue that we plan to explore in an extended journal version of this work.

---

> > ### Author Rebuttal · Reviewer_gM9i · 2026-04-03
> >
> > Thank you for the explanation.

---

### Decision · Program_Chairs · 2026-04-30

**Decision:**

Accept (spotlight)

**Comment:**

This paper proposes Bounded Log-Likelihood Loss (BLL-Loss), a simple and parameter-free alternative to standard SFT that attenuates gradient contributions from low-probability tokens, motivated by the observation that such tokens in reasoning traces often reflect realization-specific or stylistic variations rather than core reasoning. The method is theoretically well motivated through multiple derivations and gradient analysis, and is empirically validated across a wide range of mathematical, code, and general reasoning benchmarks and model scales.
Reviewers consistently find the paper well written, technically sound, and impactful, highlighting the strength of the empirical gains and the clarity of the optimization insights. While concerns were raised regarding improvement relative to related objectives (e.g., p-objective, DFT) and applicability in severe OOD settings, the rebuttal provides extensive new experiments, stronger baselines, and a clear articulation of the method’s scope and limitations.

Overall, this work offers a practical and well-substantiated training objective for reasoning tasks. I recommend Accept.